# DriveAgent-R1:
# Advancing VLM-based Autonomous Driving with Active Perception and Hybrid Thinking

**Weicheng Zheng**[1,3*]    **Xiaofei Mao**[2*]    **Nanfei Ye**[2]
**Pengxiang Li**[2]    **Kun Zhan**[2]    **Xianpeng Lang**[2]    **Hang Zhao**[1,4†]
[1]Shanghai Qi Zhi Institute    [2]LiAuto    [3]Tongji University    [4]Tsinghua University
**Project:** https://tsinghua-mars-lab.github.io/DriveAgent-R1/

## Abstract

The advent of Vision-Language Models (VLMs) has significantly advanced end-to-end autonomous driving, demonstrating powerful reasoning abilities for high-level behavior planning tasks. However, existing methods are often constrained by a passive perception paradigm, relying solely on text-based reasoning. This passivity restricts the model's capacity to actively seek crucial visual evidence when faced with uncertainty. To address this, we introduce *DriveAgent-R1*, an autonomous driving agent capable of active perception for planning. In complex scenarios, *DriveAgent-R1* proactively invokes tools to perform visual reasoning, firmly grounding its decisions in visual evidence, thereby enhancing both interpretability and reliability. Furthermore, we propose a hybrid thinking framework, inspired by human driver cognitive patterns, allowing the agent to adaptively switch between efficient text-only reasoning and robust tool-augmented visual reasoning based on scene complexity. This capability is cultivated through a three-stage progressive training strategy, featuring a core Cascaded Reinforcement Learning (Cascaded RL) phase. Extensive experiments on the Drive-Internal dataset, which is rich in long-tail scenarios, and the public nuScenes dataset show that, with only 3B parameters, *DriveAgent-R1* achieves competitive performance comparable to top closed model systems such as GPT-5 and to human driving proficiency while remaining deployment-friendly, offering a proven path toward building more intelligent autonomous driving systems.

## 1 Introduction

The paradigm of end-to-end autonomous driving has been substantially advanced by the advent of Vision-Language Models (VLMs) (Tian et al., 2024; Sima et al., 2024; Xu et al., 2024; Hu et al., 2023b; Jiang et al., 2023). By emulating human-like cognition, VLMs promise to unify perception, reasoning, and planning within a single, cohesive framework, holding the potential for superior generalization and a deeper understanding of complex scenarios. In the context of planning, the task can be decomposed into two levels: low-level motion planning and high-level behavioral decision-making (Wang et al., 2025b; Tian et al., 2024). Compared to tasking VLMs with regressing continuous physical trajectories—a task for which they are not inherently optimized—a more promising direction is to leverage their strengths in semantic understanding to predict high-level driving intentions (Wang et al., 2022; Paul et al., 2024). Accordingly, the core objective of this work is to endow agents with the capability for fine-grained, high-level behavioral planning.

To achieve this goal with greater transparency, interpretability, and robustness, we turn to the Multimodal Chain-of-Thought (M-CoT) (Xu et al., 2025a; Li et al., 2025a; OpenAI, 2025b; Cheng et al., 2025; Hu et al., 2024). The evolution of M-CoT has given rise to distinct reasoning paradigms: from early Text-based M-CoT (Xu et al., 2025a), which reasons over textual descriptions of visual inputs, to the more sophisticated, interleaved Tool-based M-CoT (Hu et al., 2024), which actively

---

*These authors contributed equally to this work.
†Corresponding at: hangzhao@mail.tsinghua.edu.cn

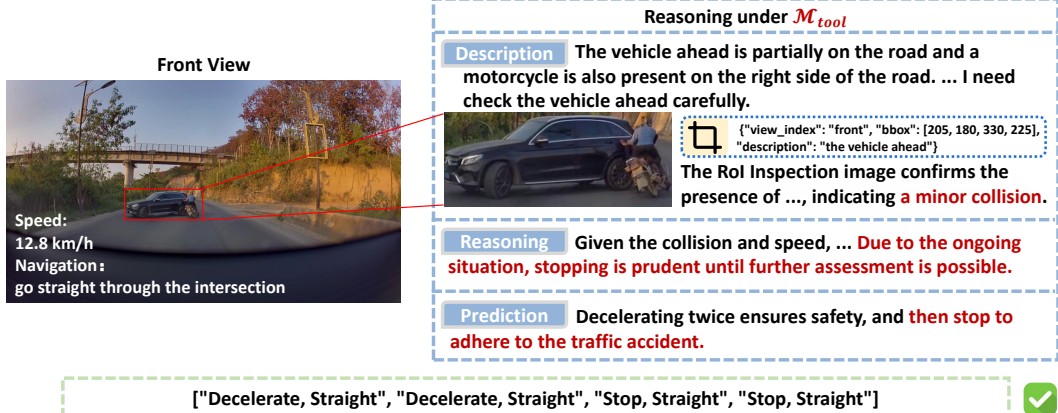

Figure 1: An illustration of *DriveAgent-R1*'s active perception capability. The agent proactively uses RoI Inspection to clarify an uncertain scene, discovering a minor collision between the vehicles ahead. This active perception corrects its initial assessment, leading to a safe plan to decelerate and then stop based on direct visual evidence. More visualization is shown in the Appendix A.10.

invokes tools to gather new visual evidence mid-reasoning, enabling a dynamic "think-while-seeing" process. While extensive research has sought to harness M-CoT for autonomous driving tasks (Tian et al., 2024; Jiang et al., 2025a; Li et al., 2025c), these efforts have predominantly remained within the confines of Text-based M-CoT, a paradigm of passive perception. The passive perception struggles with two opposing yet related challenges: (1) it cannot seek additional visual information to resolve ambiguity when the default view is insufficient; and (2) providing comprehensive multi-view data forces the model to process redundant visual inputs, increasing computational overhead and distracting it from critical cues. Human driving, by contrast, is an inherently active process of resolving uncertainty. Drivers will check blind spots or look again at a confusing traffic signal, demonstrating the ability to "actively perceive," which is the cornerstone of safe driving. Tool-based M-CoT provides a technical pathway to instantiate this more human-like driving intelligence, breaking the shackles of passive perception. However, leveraging this tool-based active perception for the core task of high-level behavioral planning remains a significant and unexplored research gap.

While this active perception paradigm is powerful, indiscriminately applying tool-based active perception to all driving scenes is computationally inefficient. Human drivers exhibit a similar cognitive efficiency, relying on intuition for simple conditions and reserving deliberate, active exploration for complex or uncertain scenarios. Such an observation suggests a more intelligent paradigm dynamically adapting its thinking mode: employing efficient Text-based M-CoT for routine cases while deploying the in-depth, Tool-based M-CoT only when the situation's complexity demands it. To our knowledge, such an adaptive mechanism that integrates these distinct M-CoT modalities for the autonomous driving planning task remains unexplored.

To bridge the identified gaps, we introduce *DriveAgent-R1*, a novel autonomous driving agent whose design is centered on two key innovations: active perception and a hybrid-thinking framework. First, it pioneers active perception for high-level behavioral planning. Leveraging a Tool-based M-CoT, *DriveAgent-R1* proactively seeks visual evidence to resolve uncertainty, grounding its decisions in verifiable perception (See Fig. 1). This is enabled by a multi-turn interactive framework centered around a specialized Vision Toolkit, which empowers the agent to acquire diverse visual information on an as-needed basis rather than passively processing redundant multi-view data (Qiao et al., 2025; Hu et al., 2023a; Sun et al., 2025). Second, we introduce a Hybrid-Thinking framework to an autonomous driving agent. It innovatively integrates efficient Text-based M-CoT with in-depth Tool-based M-CoT, allowing the agent to adaptively switch its thinking mode based on scene complexity. We cultivate this capability through a three-stage progressive training strategy. Following an initial supervised fine-tuning (SFT) stage, the agent undergoes a novel Cascaded Reinforcement Learning (Cascaded RL) phase. In this core phase, our proposed Mode-Partitioned GRPO (MP-GRPO) algorithm first strengthens each thinking mode individually and then trains the agent to adaptively select the optimal one. On the Drive-Internal, which features a diverse set of long-tail scenarios, and public nuScenes datasets, our 3B *DriveAgent-R1* achieves performance competitive with top-tier models like GPT-5 (OpenAI, 2025a) and human drivers, while maintaining deployment-friendly

efficiency. Crucially, ablation studies confirm this success is genuinely grounded in visual evidence rather than textual shortcuts, validating our active perception and hybrid-thinking framework as a path toward safer, more interpretable autonomous driving.

Our main contributions are summarized as follows:

1. We pioneer an **active perception** framework for high-level driving planning. The agent proactively invokes a vision toolkit to ground decisions in visual evidence, enhancing reasoning and reliability.

2. We introduce a **hybrid-thinking framework** that adaptively balances efficient text-only reasoning with robust, tool-augmented visual analysis. This is enabled by a novel **three-stage progressive training strategy** centered on Cascaded RL.

3. *DriveAgent-R1* achieves **both strong performance and deployment-friendly efficiency.** With only 3B parameters, *DriveAgent-R1* proves competitive with top-tier models like GPT-5 and human drivers, while significantly reducing inference latency compared to passive approaches, establishing a practical path toward scalable, intelligent autonomous driving.

## 2 DRIVEAGENT-R1

### 2.1 PROBLEM FORMULATION AND HYBRID THINKING

As illustrated in Fig. 2, the agent's primary task is to generate a long-horizon driving intention sequence from multimodal inputs. Given an initial visual context $I_0$ (from a front-view camera) and textual context $T_0$ (vehicle speed and navigation), the agent outputs an 8-second meta-action sequence $A = (a_1, a_2, a_3, a_4)$, with each action predicted at a 2-second interval. Each meta-action $a_t = (s_t, j_t)$ comprises a velocity token $s_t \in V_s$={Accelerate, Keep Speed, Decelerate, Stop} and a trajectory token $j_t \in V_j$={Straight, Right Turn, Left Turn}.

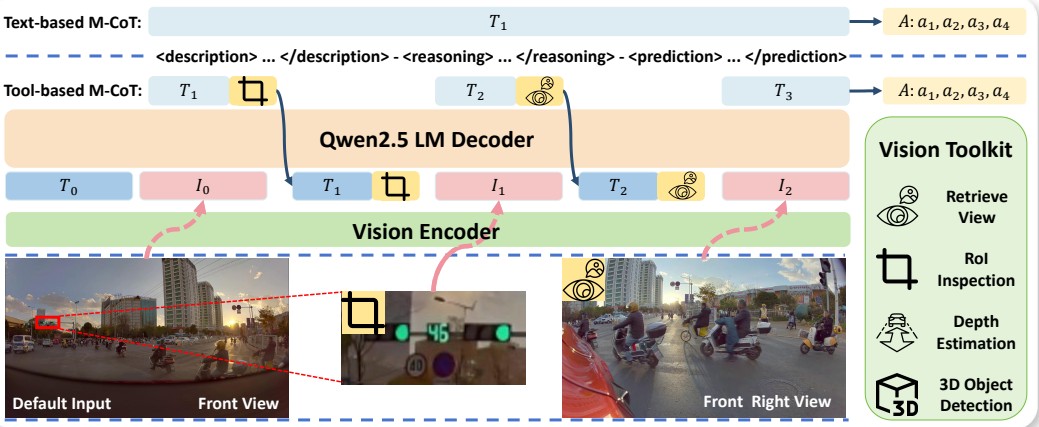

Figure 2: The Hybrid-Thinking architecture of *DriveAgent-R1*. For simple scenarios (Top), the agent uses direct text-based reasoning ($T_1 \rightarrow A$). For complex scenarios (Bottom), it iteratively interleaves thoughts ($T_k$) with tool calls to a Vision Toolkit, acquiring new visual evidence ($I_k$) to refine its decision-making. The detailed visulization of this case is shown in Fig. 10, Appendix A.10.

To achieve this, *DriveAgent-R1* employs a hybrid-thinking mechanism. It first selects a reasoning path by generating either a `<think_text>` or `<think_tool>` token. Both paths adhere to a unified CoT framework: *description*, *reasoning*, and *prediction*, where the *description* stage involves initial perception of the scene, the *reasoning* stage conducts logical analysis, and the *prediction* stage summarizes the analysis to determine the sequence.

**Text-based M-CoT** ($\mathcal{M}_{text}$). For common scenarios, *DriveAgent-R1* generates the `<think_text>` token, relying entirely on its internal knowledge and the initial input ($I_0, T_0$) to make decisions through pure text-based reasoning.

**Tool-based M-CoT ($\mathcal{M}_{\text{tool}}$).** For complex or uncertain scenarios where the initial visual context is insufficient, *DriveAgent-R1* generates the `<think_tool>` token to engage in active perception, proactively seeking visual evidence to resolve ambiguity and ground its decisions. This process updates its contextual history $H_k$ as follows:

$$H_k = H_{k-1} \oplus T_k \oplus I_k, \quad \text{for } k < K,$$

where $(T_k, \mathcal{T}_k) = \text{LMDecoder}(H_{k-1})$ and $I_k = \text{VisionEncoder}(\text{Execute}(\mathcal{T}_k))$. The process begins with an initial history $H_0 = T_0 \oplus I_0$, where $\oplus$ denotes the concatenation of token embeddings. In each step $k$, *DriveAgent-R1* generates a textual thought $T_k$ and potentially a tool call request $\mathcal{T}_k$ based on its current history $H_k$. If a tool is called, it is executed to obtain new visual information, which is then encoded into an image embedding $I_k$ and integrated into the history. The process terminates upon generating the final action sequence $A$ or reaching the maximum interaction limit $K$.

**Vision Toolkit Design.** *DriveAgent-R1* is integrated with a powerful Vision Toolkit, endowing it with the capability of active perception. These tools allow the model to explore the environment on demand during its reasoning process to acquire critical information. The toolkit includes: **(1) Retrieve View**, which fetches clear images from any camera, including historical frames from a 5-second memory pool; **(2) RoI Inspection**, a "zoom-in" function that crops and magnifies a specified Region of Interest (RoI) from a higher-resolution image to provide rich visual details; **(3) Depth Estimation**, which generates a depth map to provide crucial 3D spatial awareness; **(4) 3D Object Detection**, an open-vocabulary tool that identifies and localizes objects in 3D space. Detailed descriptions and tool call formats are provided in Appendix A.2.

## 2.2 Autonomous Driving Domain Alignment

Recent studies (Li et al., 2025c; Chen et al., 2025; Wang et al., 2025a) indicate that general VLMs exhibit a shortcut tendency in driving planning: they rely on low-dimensional textual cues and neglect high-dimensional visual input. To first develop visual sensitivity and ensure decisions are grounded in evidence, inspired by Drive-R1 (Li et al., 2025c), we perform autonomous-driving domain alignment on the base model before high-level behavioral planning training. To this end, we build an autonomous-driving VQA dataset from images collected in real traffic, totaling 530K question–answer pairs. Questions in the dataset fall into four main categories: (i) scene description, (ii) recognition of traffic entities, (iii) localization of critical targets, and (iv) traffic commonsense and rules. Using this dataset, we fully fine-tune all parameters of Qwen2.5-VL-3B (Bai et al., 2025) and obtain *DriveAlign-3B*, a model highly sensitive to visual evidence in driving scenes. *DriveAlign-3B* is used as the unified initialization for the subsequent three-stage training, laying the visual-alignment foundation for *DriveAgent-R1*'s active perception and hybrid thinking. See the Appendix A.3 for details of the VQA data construction.

## 2.3 Progressive Training via SFT and Cascaded RL

To cultivate the hybrid-thinking capabilities of *DriveAgent-R1*, we devise a progressive training strategy composed of an initial Supervised Fine-Tuning (SFT) stage and a subsequent two-stage Cascaded RL phase shown in Fig. 3. This overall strategy follows a "foundation building → mode strengthening → intelligent selection" paradigm.

### 2.3.1 Stage 1: Dual-Mode Supervised Fine-Tuning

We begin by cold-starting the model using SFT to endow it with a foundational understanding of the format and semantic boundaries of both thinking modes. To this end, we construct a high-quality dataset via a three-stage automated pipeline. The pipeline first partitions raw data into a tool-unnecessary set $D_{\text{text}}$ and a tool-necessary set $D_{\text{tool}}$. $D_{\text{text}}$ comprises simple scenarios where Qwen2.5-VL-3B already achieves high accuracy without tools, indicating no need for active perception. Conversely, $D_{\text{tool}}$ includes complex cases where the powerful Qwen2.5-VL-72B shows a performance gain only when using tools, establishing a criterion for tool necessity. Next, Qwen2.5-VL-72B generates mode-specific CoT annotations, aligning with recent findings (Zhou et al., 2025) that demonstrate the efficacy of constrained or prompted generation for high-quality annotation. To ensure data quality, these annotations undergo a rigorous validation and filtering process: a separate judge model scores the reasoning coherence, and samples falling below a quality threshold are automatically

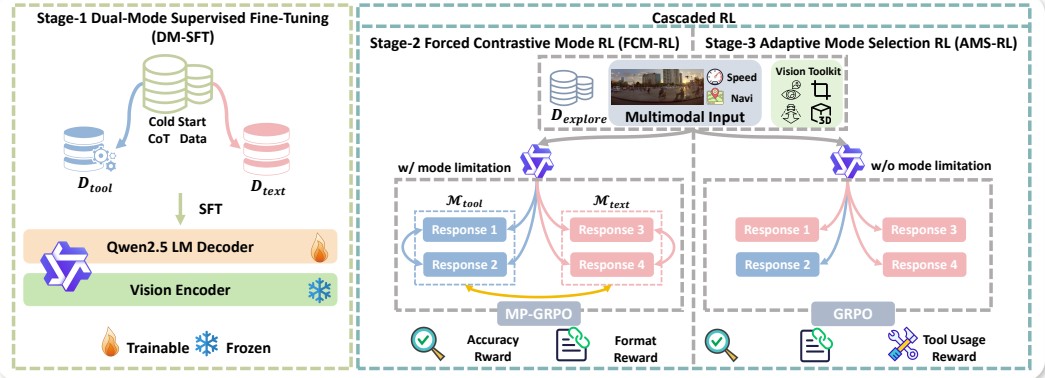

Figure 3: The progressive three-stage training strategy for *DriveAgent-R1*. The process begins with (1) DM-SFT to establish a foundational understanding of both thinking modes. This is followed by a core Cascaded RL phase, where (2) FCM-RL strengthens each mode independently, and (3) AMS-RL trains the agent to adaptively select the optimal mode.

regenerated. Finally, the data is passed through rule-based cleaning to create a balanced, high-quality dataset for SFT. A complete description of this pipeline is provided in Appendix A.4.

### 2.3.2 STAGE 2: FORCED CONTRASTIVE MODE REINFORCEMENT LEARNING

The first stage of our Cascaded RL phase aims to deepen the agent's independent performance in each mode. We adopt Group Relative Policy Optimization (GRPO) (Guo et al., 2025) as our foundational algorithm, which has proven effective in multimodal reasoning and autonomous driving (Jiang et al., 2025a; Liu et al., 2025; Su et al., 2025; Zheng et al., 2025). Unlike PPO, GRPO eliminates the need for a critic model by estimating the baseline from a group of outputs. Specifically, for each query $q$, GRPO samples a group of outputs $\{o_1, o_2, \ldots, o_G\}$ from the old policy $\pi_{\theta_{old}}$ and optimizes the policy $\pi_\theta$ by maximizing the following objective:

$$\mathcal{J}_{GRPO}(\theta) = \mathbb{E}_{q \sim P(Q), \{o_i\} \sim \pi_{\theta_{old}}} \left[ \frac{1}{G} \sum_{i=1}^{G} \left( \min\left(w_i A_i, \text{clip}(w_i, 1 \pm \epsilon) A_i\right) - \beta \mathbb{D}_{KL}(\pi_\theta || \pi_{ref}) \right) \right],$$

where $w_i = \frac{\pi_\theta(o_i|q)}{\pi_{\theta_{old}}(o_i|q)}$ is the importance sampling ratio, $\epsilon$ and $\beta$ are hyper-parameters, and the advantage $A_i$ is computed by normalizing the rewards within the group. Building upon this standard formulation, we propose Mode-Partitioned GRPO (MP-GRPO) to prevent the agent from developing a bias against an initially weaker thinking mode. As illustrated in Fig. 3, the core mechanism of MP-GRPO is to enforce balanced exploration across both thinking modes. For each input $q$ from $D_{\text{explore}}$, we compel the agent to generate two mode-specific sets of $G/2$ responses by forcing the corresponding mode token (<think_text> or <think_tool>). This creates a full group of $G$ responses:

$$\mathcal{O}(q) = \{o_i^{\text{text}}\}_{i=1}^{G/2} \cup \{o_j^{\text{tool}}\}_{j=1}^{G/2},$$

where each individual response, denoted by $o$, is sampled from the old policy $\pi_{\theta_{old}}$. Crucially, all $G$ responses are then aggregated into this unified set $\mathcal{O}(q)$ for reward normalization and advantage calculation. This design creates a **multi-dimensional contrastive learning signal**: it facilitates not only **intra-mode** comparison between different response trajectories but also **inter-mode** comparison. This multi-faceted contrast allows the agent to simultaneously enhance its specific capabilities in both modes and gain an initial understanding of the suitability of each mode for different scenarios, laying a critical foundation for the final stage of adaptive selection.

**Reward Design.** The agent's learning is guided by an outcome-driven reward function that evaluates the entire reasoning trajectory. This reward is a composite of two key components: an accuracy reward ($R_{\text{acc}}$) and a format consistency reward ($R_{\text{fmt}}$), formulated as follows:

$$R = R_{\text{acc}} + R_{\text{fmt}},$$

where the accuracy reward $R_{\text{acc}}$ is a weighted Levenshtein distance against the ground-truth sequence, with position-action weights to counteract action imbalance (see Appendix A.5 for details). The

format consistency reward $R_{\text{fmt}}$ penalizes structural errors and incorrect mode usage, such as invoking tools in the $\mathcal{M}_{\text{text}}$ mode.

### 2.3.3 STAGE 3: ADAPTIVE MODE SELECTION REINFORCEMENT LEARNING

The final stage of Cascaded RL cultivates true hybrid thinking by training the agent to intelligently select the optimal mode. Using the native GRPO algorithm (Guo et al., 2025), the agent learns to generate the initial mode-selection token itself and commit to the corresponding reasoning path.

**Reward Design.** To guide this learning process, the reward function is augmented with a conditional tool-usage reward, $R_{\text{tool}}$, which encourages the effective and judicious use of tools only when the $\mathcal{M}_{\text{tool}}$ mode is chosen. The total reward is defined as:

$$R = R_{\text{acc}} + R_{\text{fmt}} + \mathbb{I}(\text{mode} = \mathcal{M}_{\text{tool}}) \cdot R_{\text{tool}},$$

where the definitions of $R_{\text{acc}}$ and $R_{\text{fmt}}$ remain consistent with Stage 2. The core of $R_{\text{tool}}$ is a **contrastive mechanism** that promotes impactful tool use. Within each group of $G$ generated responses, we establish a baseline by calculating the average accuracy of all text-only trajectories ($\bar{Acc}_{\text{text}}$). A tool-based trajectory is rewarded only if its own accuracy ($R_{\text{acc}}$) surpasses this baseline by a margin, i.e., $R_{\text{tool}} \propto (R_{\text{acc}} - \bar{Acc}_{\text{text}} - \text{margin})$. This design explicitly penalizes superfluous tool calls, incentivizing the agent to invoke active perception only when it provides a clear performance advantage over pure text-based reasoning. A detailed formulation is provided in the Appendix A.5.

## 3 EXPERIMENTS

### 3.1 SETUP

#### 3.1.1 DATASET

**Automated Label Generation for Meta-actions.** We produce labels for meta-action sequences with a rule-first, VLM-corrected pipeline. Starting from vehicle trajectory points, we use a 3-second sliding window to map each segment to a predefined vocabulary of speed and trajectory actions, employing thresholds iteratively optimized via dual VLM-human feedback. This yields a four-step discrete meta-action sequence for the next 8 seconds at 2-second intervals. To correct a small number of ambiguous or boundary cases, we then feed the resulting sequence and its BEV-map visualization into GPT-4.1 for refinement. Full rules and optimization details are in the Appendix A.7.

**Training Data.** All training data are drawn from our Drive-Internal dataset, which comprises 35K video clips specifically targeting long-tail and complex scenarios. This dataset features over 40 distinct driving situations, including challenging events like road works and animal crossings. To leverage its rich diversity, one or two key frames are sampled from each clip, focusing on the moments surrounding critical events. For Stage 1 (DM-SFT), we generate 4K high-quality CoT data using the pipeline from Sec. 2.3.1, splitting it evenly into the tool-necessary set $D_{\text{tool}}$ and the tool-unnecessary set $D_{\text{text}}$ (2K each). For the Cascaded RL phase, we construct an exploratory set $D_{explore}$ by sampling 15K instances from the data not assigned to $D_{tool}$ or $D_{text}$, ensuring the sampling maintains a balanced distribution of actions at each timestep in the sequence.

**Test Data.** We randomly sample 1.5K instances from Drive-Internal as the in-distribution test set Drive-Internal$_{\text{test}}$. To assess generalization, we also sample 2K instances from the nuScenes validation set as nuScenes$_{\text{test}}$ for cross-dataset testing. All nuScenes samples are taken from the middle of each clip to ensure sufficient history and future trajectories. Meta-action labels on nuScenes are generated by the same pipeline.

#### 3.1.2 EVALUATION METRICS

**Accuracy (Acc).** We evaluate planning precision using First-Frame and Sequence Average Joint Accuracy. A prediction is considered correct only if the composite meta-action (both speed and trajectory) exactly matches the ground truth. To encourage safer behavior, we also employ a Relaxed Speed Matching mechanism, awarding partial scores (0.5 and 0.2) for predictions that are one and two levels safer than the ground truth, respectively.

Table 1: Results on the Drive-Internal$_{test}$ and nuScenes$_{test}$. All models are evaluated using one-shot prompting under two conditions: without tools and with access to the visual toolkit. To ensure a consistent, structured reasoning format, closed-source models were queried via their official APIs in a "no-thinking" mode. Parentheses show the absolute gain from using tools.

| Model | Drive-Internal$_{test}$ | | | | nuScenes$_{test}$ | | | |
|---|---|---|---|---|---|---|---|---|
| | First-Frame Joint Acc. (%) | | Seq. Avg. Joint Acc. (%) | | First-Frame Joint Acc. (%) | | Seq. Avg. Joint Acc. (%) | |
| | w/o Tools | w/ Tools | w/o Tools | w/ Tools | w/o Tools | w/ Tools | w/o Tools | w/ Tools |
| Human | 49.59 | | 49.29 | | 50.48 | | 48.24 | |
| Qwen2.5-VL-3B | 24.06 | 23.64 (-0.42) | 24.98 | 22.63 (-2.35) | 30.18 | 28.17 (-2.01) | 23.48 | 21.58 (-1.90) |
| Qwen2.5-VL-7B | 27.58 | 24.01 (-3.57) | 29.84 | 24.98 (-4.86) | 33.11 | 32.84 (-0.27) | 27.60 | 28.82 (+1.22) |
| Qwen2.5-VL-72B | 32.76 | 32.97 (+0.21) | 38.80 | 39.61 (+0.81) | 43.26 | 43.87 (+0.61) | 39.13 | 40.47 (+1.34) |
| Doubao-Seed-1.6 | 34.92 | 38.98 (+4.06) | 40.22 | 41.37 (+1.15) | 45.63 | 46.29 (+0.66) | 41.91 | 43.10 (+1.19) |
| Gemini-2.5-Flash | 41.83 | 43.38 (+1.55) | 42.14 | 43.42 (+1.28) | 45.46 | 46.60 (+1.14) | 42.69 | 44.07 (+1.38) |
| GPT-4.1 | 39.99 | 43.18 (+3.19) | 42.14 | 43.43 (+1.29) | 46.84 | 48.25 (+1.41) | 43.63 | 44.72 (+1.09) |
| GPT-5 | 56.30 | **56.48** (+0.18) | 47.19 | **47.97** (+0.78) | 48.75 | 49.11 (+0.36) | 44.85 | 45.14 (+0.29) |
| DriveAgent-R1 | 45.27 | 51.34 (+6.07) | 43.29 | 45.42 (+2.13) | 52.58 | **52.96** (+0.38) | 44.43 | **47.10** (+2.67) |

**Mode Selection Accuracy (MSA).** To quantify *DriveAgent-R1*'s adaptive mode selection, we introduce the Mode Selection Accuracy (MSA), inspired by Jiang et al. (2025b). It measures the alignment between the agent's adaptively chosen mode $m_{adaptive}$ and the optimal mode $m_i^*$. The optimal mode $m_i^*$ is defined as whichever mode ($\mathcal{M}_{tool}$ or $\mathcal{M}_{text}$) achieves higher accuracy for a given sample $q_i$. MSA is then calculated as:

$$MSA = \frac{1}{N} \sum_{i=1}^{N} \mathbb{I}(m_{adaptive}(q_i) = m_i^*)$$

### 3.1.3 IMPLEMENTATION DETAILS

All training is conducted on 8 H20 GPUs. For all multimodal inputs in our experiments, the maximum number of pixels per image is set to 259,200. Unless specified, only the front-view camera image serves as the default visual input. During the Cascaded RL phase, we set the sampling temperature to 1.0, limit the maximum number of tool calls to 3, and group size to 4 per sample. For all evaluations, the sampling temperature is set to 0.7.

## 3.2 MAIN RESULTS

### 3.2.1 COMPARISON WITH SOTA VLMS AND HUMAN DRIVERS

To comprehensively evaluate *DriveAgent-R1*, we benchmark it against a suite of state-of-the-art closed-source VLMs, the open-source Qwen2.5-VL series, and a human performance baseline on both the Drive-Internal$_{test}$ and nuScenes$_{test}$. More details are shown in Appendix A.8. The main results are summarized in Table 1, leading to two key observations.

***DriveAgent-R1 achieves performance comparable to GPT-5 and human drivers with only 3B parameters.*** Despite its 3B parameter size, *DriveAgent-R1* shows remarkable performance and the most effective tool use, achieving a +6.0% accuracy gain on Drive-Internal$_{test}$. Its performance is competitive with both GPT-5 and human drivers. Notably, on the nuScenes$_{test}$, *DriveAgent-R1* surpasses GPT-5 in sequence accuracy (47.10% vs. 45.14%) and approaches human-level proficiency.

***Visual tool is a double-edged sword.*** While VLMs like GPT-4.1 and Gemini-2.5-Flash improved with tool access, the Qwen2.5-VL-3B and -7B models showed consistent performance degradation. This indicates that effective tool use is a non-trivial skill, requiring either a high level of base model capability or targeted, specialized training to master.

### 3.2.2 PERFORMANCE ON DRIVEBENCH

To further validate DriveAgent-R1 against existing state-of-the-art VLM-based driving agents, we conducted an evaluation on DriveBench (Xie et al., 2025), a benchmark designed for high-level driving tasks. We compare our model with DriveLM (Sima et al., 2024) and Dolphins (Ma et al., 2024).

Table 2: **Performance Comparison on DriveBench.** We compare DriveAgent-R1 with representative VLM-based driving agents.

| Method | Perception | Prediction | Planning | Behavior |
|---|---|---|---|---|
| DriveLM | 16.85 | **44.33** | **68.71** | 42.78 |
| Dolphins | 9.59 | 32.66 | 52.91 | 18.81 |
| **DriveAgent-R1 (Ours)** | **34.07** | 32.85 | 61.89 | **43.69** |

Table 3: **Open-loop planning performance on nuScenes validation set.** We report the L2 Displacement Error (DE) and Collision Rate (CR) at different time horizons. **Bold** and underlined denote the best and second-best results, respectively.

| Model | ADE (m) ↓ | | | | Collision Rate (%) ↓ | | | |
|---|---|---|---|---|---|---|---|---|
| | 1s | 2s | 3s | avg | 1s | 2s | 3s | avg |
| UniAD | 0.44 | 0.67 | 0.96 | 0.69 | 0.04 | 0.08 | 0.23 | 0.12 |
| VAD-Base | 0.17 | 0.34 | 0.60 | 0.37 | 0.07 | 0.10 | 0.24 | 0.14 |
| EMMA | 0.14 | 0.29 | 0.54 | 0.32 | - | - | - | - |
| DriveVLM-Dual | 0.15 | 0.29 | 0.48 | 0.31 | 0.05 | 0.08 | 0.17 | 0.1 |
| OmniDrive | 0.14 | 0.29 | 0.55 | 0.33 | 0.00 | 0.13 | 0.78 | 0.3 |
| OpenDriveVLA (3B) | 0.14 | 0.30 | 0.55 | 0.33 | 0.02 | 0.07 | 0.22 | 0.1 |
| **DriveAgent-R1 (Ours)** | **0.13** | **0.24** | **0.47** | **0.28** | 0.01 | 0.12 | 0.30 | 0.14 |

***DriveAgent-R1 achieves superior scene understanding and decision-making.*** As shown in Table 2,, DriveAgent-R1 achieves a Perception score of 34.07, doubling DriveLM's 16.85. This substantial margin confirms that proactive tool invocation captures critical visual details often missed by passive paradigms. Moreover, our top-ranking Behavior score (43.69) demonstrates that the DriveAgent-R1 effectively translates this rich visual evidence into safe, correct decision-making.

## 3.3 PERFORMANCE OF MOTION PLANING

To validate whether the high-level reasoning of DriveAgent-R1 translates into effective low-level control, we extended the architecture with a lightweight MLP motion planning head to regress physical trajectories on the nuScenes dataset. The motion head acts as an "executor" for the frozen DriveAgent-R1. It uses a simple MLP taking three inputs: meta-action sequence (one-hot), visual tokens, and ego-status, predicting 3s future waypoints at 2Hz.

***High-Level Reasoning Enables Precise and Safe Planning.*** On the nuScenes validation set, DriveAgent-R1 achieves an average ADE of 0.28m, outperforming strong baselines like DriveVLM-Dual ($0.31m$) while maintaining a comparable collision rate. This confirms that the reasoning-driven meta-actions are the decisive factor enabling effective low-level control.

Table 4: **Analysis of Foundational Capabilities.** Overall scores on domain-specific and 8 general VLM benchmarks. Numbers in blue denote the improvement over Qwen2.5-VL-3B. Detail results are in Appendix A.9.1.

| Model | DriveAlign-VQA$_{test}$ | General Benchmarks |
|---|---|---|
| Qwen2.5-VL-3B | 72.9 | 64.4 |
| Qwen2.5-VL-7B | 77.0 | 68.9 |
| **DriveAlign-3B** | **84.6** (+11.7) | **66.4** (+2.0) |

Table 5: **Impact of Domain Alignment on High-Level Behavioral Planning Task.** We report sequence accuracy on Drive-Internal$_{test}$. The relative performance drop (%) upon removing images is shown in parentheses.

| Base Model | Model | Seq. Avg. Joint Acc. (%) | |
|---|---|---|---|
| | | w/ image | w/o image (rel. drop) |
| Qwen2.5-VL-3B | DriveAgent-R1-Variant | 43.56 | 38.79 (−11.0%) |
| DriveAlign-3B | DriveAgent-R1 | **45.42** | 38.24 (−15.8%) |

## 3.4 ABLATION ON DOMAIN ALIGNMENT

To validate our domain alignment strategy, we analyze its impact on both the base model's capabilities and the downstream high-level behavioral planning task.

***Domain Alignment Enhances Driving-Specific and General Capabilities.*** As shown in Table 4, domain alignment significantly boosts performance on our driving-specific VQA benchmark (+11.7) and also improves scores across a suite of general VLM benchmarks (+2.0). This confirms our strategy successfully injects domain knowledge while strengthening general capabilities.

***Domain Alignment Boosts Planning Performance and Mitigates Visual Neglect.*** As shown in Table 5, the agent built on our aligned model achieves higher accuracy (+1.86 Sequence Avg-Joint Acc). Crucially, its performance drops more significantly when images are removed (-15.8% vs. -11.0%), confirming its gains are grounded in visual evidence rather than textual shortcuts, thereby mitigating the visual neglect problem.

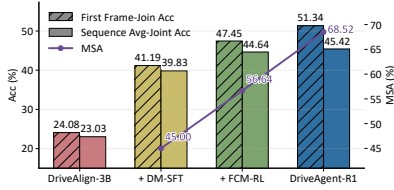

Figure 4: **Progressive training gains on Drive-Internal$_{\text{test}}$.** Accuracy in $\mathcal{M}_{\text{adaptive}}$ mode and MSA improve with each training stage.

Table 6: **Ablation on the progressive training strategy on the Drive-Internal$_{\text{test}}$.** We evaluate different combinations of our three training stages. To ensure fair comparison, we also test variants where a single RL stage is trained for two epochs, matching the total RL epochs of *DriveAgent-R1*.

| Variant | Training Stage | | | Seq. Avg. Joint Acc. (%) | | | MSA (%) |
|---|---|---|---|---|---|---|---|
| | DM-SFT | FCM-RL | AMS-RL | $\mathcal{M}_{\text{tool}}$ | $\mathcal{M}_{\text{text}}$ | $\mathcal{M}_{\text{adaptive}}$ | |
| Variant-1 (SFT Only) | 1 epoch | - | - | 42.02 | 40.65 | 40.88 | 45.00 |
| Variant-2 (+FCM) | 1 epoch | 1 epoch | - | 44.30 | 42.20 | 44.64 | 56.64 |
| Variant-3 (+FCM x2) | 1 epoch | 2 epochs | - | 43.89 | 42.81 | 44.91 | 57.32 |
| Variant-4 (+AMS) | 1 epoch | - | 1 epoch | 43.78 | 41.64 | 43.43 | 57.55 |
| Variant-5 (+AMS x2) | 1 epoch | - | 2 epochs | 44.76 | 42.56 | 44.13 | 61.61 |
| **DriveAgent-R1 (FCM→AMS)** | **1 epoch** | **1 epoch** | **1 epoch** | **44.97** | **43.29** | **45.42** | **68.52** |

## 3.5 ABLATION ON PROGRESSIVE TRAINING STRATEGY

To validate our progressive training strategy, we ablate different stage combinations in forced-text ($\mathcal{M}_{\text{text}}$), forced-tool ($\mathcal{M}_{\text{tool}}$), and adaptive-selection ($\mathcal{M}_{\text{adaptive}}$) modes on both the Drive-Internal$_{\text{test}}$ and nuScenes$_{\text{test}}$ datasets to confirm the method's effectiveness (full nuScenes$_{\text{test}}$ results are in the Appendix A.9.2). Our analysis yields two key findings.

***Progressive Training Yields Stable Performance Gains.*** As illustrated in Fig. 4, our strategy achieves consistent performance improvements, resulting in a steady increase in both accuracy and MSA.

***The Cascaded RL Strategy Surpasses Single-Strategy Stacking due to Functional Complementarity.*** Table 6 reveals that our cascaded RL strategy is synergistic, outperforming single-strategy stacking with the same total training epochs (Variant-3/5). This superiority stems from the complementary roles of its two stages: FCM-RL sharpens execution proficiency (achieving higher forced-mode accuracies of 44.30% vs. 43.78% in $\mathcal{M}_{\text{tool}}$ and 42.20% vs. 41.64% in $\mathcal{M}_{\text{text}}$, while AMS-RL hones selection intelligence (yielding a higher MSA of 57.55% vs. 56.64%). Our cascaded approach successfully integrates these distinct strengths for superior overall performance.

Table 7: **Ablation on Active vs. Passive Perception.** We compare *DriveAgent-R1* with passive baselines on Drive-Internal$_{\text{test}}$. Relative drop (%) without images in parentheses.

| Model | Seq. Avg. Joint Acc. (%) | |
|---|---|---|
| | w/ image | w/o image (rel. drop) |
| Passive-FV | 42.70 | 39.64 (-7.2%) |
| Passive-SV | 43.10 | 39.01 (-9.5%) |
| DriveAgent-R1 (Retrieve View Only) | 44.39 | 38.78 (-12.6%) |
| **DriveAgent-R1 (Full Toolkit)** | **45.42** | 38.24 (-15.8%) |

Table 8: **Performance and Efficiency Analysis.** We compare inference latency and average output tokens for different perception and reasoning strategies.

| Model/Mode | Latency (s) | Avg. Token |
|---|---|---|
| Passive-SV | 8.50 | 210.77 |
| DriveAgent-R1 ($\mathcal{M}_{\text{tool}}$) | 7.91 | 314.45 |
| **DriveAgent-R1 ($\mathcal{M}_{\text{adaptive}}$)** | 6.74 | 265.57 |

## 3.6 ABLATION ON ACTIVE VERSUS PASSIVE PERCEPTION

To isolate the benefits of active perception, we compare *DriveAgent-R1* against two passive perception baselines: Passive-FV (front-view) and Passive-SV (surrounding views), all of which were trained with the same SFT and RL pipeline for a fair comparison. For a controlled comparison against Passive-SV, we evaluate a restricted variant, ***DriveAgent-R1* (Retrieve View Only)**, which is limited to using only the "Retrieve View" tool. As shown in Table 7, our findings are twofold.

***Active Selection is Superior to Passive Input.*** Active perception, as demonstrated by *DriveAgent-R1* (Retrieve View Only), shows a dual advantage. First, it overcomes information deficits (outperforming Passive-FV by +1.69 points). Second, it mitigates the redundancy of multiple fixed views. Despite sharing an identical information ceiling (the same pool of six views), *DriveAgent-R1* outperforms

the Passive-SV by +1.29 points. This confirms that actively retrieving task-relevant information on demand is more effective than passively processing a fixed set of inputs.

***Active Perception Deepens Visual Reliance.*** A clear trend emerges: the more active and capable the agent's perception, the more its performance drops when visual inputs are removed. This confirms our framework successfully fosters a genuine, vision-driven decision-making process, steering the model away from textual shortcuts. This heightened dependence on vision, however, also underscores the criticality of correctly interpreting the acquired evidence. In complex scenarios, misinterpreting new visual information can lead to errors, a challenge we explore in Appendix A.11.

### 3.7 EFFICIENCY ANALYSIS

To assess the computational efficiency of our framework, we evaluate inference latency and average output token length. Latency is measured on a single H20 GPU with a batch size of 1 using the vLLM (Kwon et al., 2023) inference framework, averaged over 100 samples and repeated across 5 runs. The results, summarized in Table 8, highlight two key advantages of our design.

***Active Perception is More Efficient than Passive Input.*** By selectively retrieving visual data only when needed, *DriveAgent-R1* avoids the computational overhead of processing the redundant images required by Passive-SV. This targeted approach reduces inference latency by over 20%.

***Hybrid Thinking Balances Performance and Cost.*** The advantage of hybrid thinking is clear when comparing the ($\mathcal{M}_{\text{adaptive}}$ and $\mathcal{M}_{\text{tool}}$ modes. *DriveAgent-R1* under $\mathcal{M}_{\text{adaptive}}$ reduces both inference latency (from 7.91s to 6.74s) and output tokens (from 314.45 to 265.57).

## 4 RELATED WORK

**VLMs for Autonomous Driving.** Recent advancements have seen Vision-Language Models (VLMs) applied to autonomous driving, unifying perception, reasoning, and planning (Sima et al., 2024; Tian et al., 2024; Mao et al., 2023). Research has primarily followed two paths: enhancing structured scene understanding (Sima et al., 2024; Qian et al., 2025; Nie et al., 2024; Shao et al., 2024) and translating reasoning into actionable behaviors, from high-level decisions to low-level motion planning (Jiang et al., 2025a; Li et al., 2025c; Tian et al., 2024). However, existing VLM-based planners predominantly operate under a passive perception paradigm, reasoning over a fixed set of inputs. *DriveAgent-R1* breaks from this by introducing an active perception framework, where the agent proactively invokes a visual toolkit to ground its decisions in visual evidence.

**Multimodal Reasoning with Chain-of-Thought.** Chain-of-Thought (CoT) has significantly improved the reasoning capabilities of LLMs (Wei et al., 2022; Chu et al., 2023; Wang & Zhou, 2024) and has been extended to the multimodal domain (M-CoT) (Xu et al., 2025a; He et al., 2024; Wang et al., 2024; Mondal et al., 2024; Liu et al., 2025). The evolution of M-CoT has progressed from Text-based methods, which reason over static textual descriptions (Xu et al., 2025a; Zhao et al., 2025; Li et al., 2025b), to more dynamic, interleaved approaches. Among these, Tool-based M-CoT is a promising direction, enabling models to actively seek new information during reasoning (Qian et al., 2025; OpenAI, 2025b; Hu et al., 2024; Su et al., 2025; Zheng et al., 2025). Our work advances this frontier by introducing a novel hybrid-thinking paradigm that adaptively integrates efficient Text-based M-CoT with robust Tool-based M-CoT, allowing the agent to modulate its reasoning depth according to the scene's complexity.

## 5 CONCLUSION AND LIMITATIONS

We introduced *DriveAgent-R1*, an autonomous driving agent that pioneers active perception and hybrid thinking for high-level behavioral planning. Our 3B model achieves performance competitive with top-tier systems like GPT-5 and human drivers while being deployment-friendly. However, a key limitation is the agent's over-reliance on new visual evidence from its tools, as detailed in our failure case analysis (Appendix A.11). Future work will directly address this challenge through two complementary directions: (1) strengthening the agent's foundational understanding of complex road topologies and traffic rules, and (2) cultivating a more nuanced reasoning process capable of resolving informational conflicts.

## ACKNOWLEDGMENTS

This work was sponsored by the Beijing Nova Program.

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

# A APPENDIX

## A.1 USAGE OF LLM

Large Language Models were utilized as a writing aid in the preparation of this manuscript. Their use was strictly limited to improving the clarity and readability of the text through sentence polishing and grammar correction. Additionally, we consulted LLMs for brainstorming potential names for concepts and components, such as $\mathcal{M}_{adaptive}$, DriveAlign-3B, and Drive-Internal. The core methodology, experimental results, and scientific insights presented in this paper are the original work of the authors.

## A.2 VISION TOOLKIT: DETAILED DESCRIPTIONS AND CALL FORMATS

**Retrieve View.** This tool allows the agent to request the image from any specific viewpoint. Its design is motivated by human drivers who consciously check specific mirrors or windows to get a clear view of the road conditions before critical maneuvers like lane changes or turns. More importantly, this tool incorporates a Historical Memory Pool, which caches images from all viewpoints over the past 5 seconds, sampled at 1Hz. This enables the model to intelligently look back at historical frames to assess changes in traffic light status or the movement trends of dynamic objects, thus avoiding the high computational cost and latency associated with processing redundant information from full video sequences.

```
<tool_call>
    <tool_name>Retrieve View</tool_name>
    <params>{"frame_index": "-1s", "view_index": "front_left"}</params>
</tool_call>
```

Listing 1: Call Format of Retrieve View

**RoI Inspection.** This tool empowers the agent with a "zoom-in" capability for meticulous examination of critical areas. Based on a selected view, the agent can request to retrieve a higher-resolution version of the image and then crop and magnify a specific Region of Interest (RoI) by providing the absolute pixel coordinates of its bounding box. This is not merely a passive utility; it is a direct manifestation of the model's Inherent Localization Ability. This allows for fine-grained analysis and focused viewing of any region the agent deems important, which is vital for confirming details such as the status of distant traffic lights or the text on traffic signs.

```
<tool_call>
    <tool_name>RoI Inspection</tool_name>
    <params>{"view_index": "front_left", "bbox": [x_min, y_min, x_max,
    y_max], "description": "the traffic lights"}</params>
</tool_call>
```

Listing 2: Call Format of RoI Inspection

**Depth Estimation.** Accurately perceiving the 3D spatial relationship between the ego-vehicle and other traffic participants (e.g., vehicles, pedestrians) is fundamental to safe driving. This tool leverages the state-of-the-art monocular depth estimation algorithm, Depth Anything V2 (Yang et al., 2024), to generate a depth map for a specified view. The resulting depth map provides rich geometric information, enabling the model to intuitively grasp the relative distances and spatial layout of objects, as well as the overall road structure, which is crucial for path planning and obstacle avoidance.

```
<tool_call>
    <tool_name>Depth Estimation</tool_name>
    <params>{"view_index": "front"}</params>
</tool_call>
```

Listing 3: Call Format of Depth Estimation

**3D Object Detection.** To comprehensively understand the dynamic environment, the agent must identify key objects and ascertain their precise 3D spatial information (position, size, and orientation). We integrate an advanced open-vocabulary, single-monocular 3D object detection tool that works with arbitrary camera views (Zhang et al., 2025). This tool not only detects a predefined set of common objects like cars, pedestrians, and traffic signs but its "open-vocabulary" nature also allows the model to dynamically specify new objects for detection based on the current scene's needs, demonstrating remarkable flexibility and adaptability. The output from this tool provides indispensable input for subsequent trajectory prediction and risk assessment.

```
<tool_call>
    <tool_name>3D Object Detection</tool_name>
    <params>{"view_index": "front", "object_text": "barrier"}</params>
</tool_call>
```

Listing 4: Call Format of 3D Object Detection

## A.3 DATASET CONSTRUCTION

### A.3.1 DRIVEALIGN-VQA DATASET CONSTRUCTION

To build a strong visual foundation for *DriveAgent-R1*, we constructed the DriveAlign-VQA dataset, a large-scale, high-quality collection of question-answer pairs specific to autonomous driving. The construction followed a systematic generation-and-validation pipeline, ensuring data diversity, relevance, and accuracy. The key stages are outlined below:

1. **Image Collection and Diversification.** We first collected 1 million frames from real-world driving scenarios. To ensure a wide variety of scenes, we converted these images into CLIP embeddings, performed feature-based clustering, and then sampled a fixed number of images from each cluster. This process guarantees a diverse representation of road types, weather conditions, and traffic densities.

2. **QA Pair Generation.** We employed the powerful Qwen2.5-VL-72B model to generate question-answer (QA) pairs from the curated images. The generation process used a mixed strategy: 70% of questions were derived from predefined templates covering key driving-related topics, while the remaining 30% were generated freely to capture more nuanced scene details (see the prompt below). The generation randomly utilized either a single front-view image or all six camera views to encourage robust, multi-perspective understanding.

3. **Quality Validation and Filtering.** To ensure the highest quality, we used Doubao-Seed-1.6 as an automated "judge" model to validate every generated QA pair. The judge model scored each pair on dimensions such as question relevance, answer correctness, and grounding in visual evidence. Only pairs that received a perfect score were retained for the next stage.

4. **Diversity Enhancement.** In the final step, we utilized Doubao-Seed-1.6 again to rewrite the validated questions. This step was crucial for increasing linguistic diversity and preventing the model from overfitting to specific phrasings in the templates. During this phase, we also filtered out overly simple or trivial QA pairs to maintain a high level of complexity in the final dataset.

This rigorous pipeline ultimately yielded the DriveAlign-VQA dataset, comprising 530K high-quality question-answer pairs that effectively develop driving-specific visual sensitivity in our base model.

---

**Prompt for Free-Form VQA Generation**

I am preparing high-quality visual question answering (VQA) training data to pretrain a Qwen2.5-VL-3B vision-language model (VLM) for autonomous driving scene understanding. For each input front-view driving scene image I provide, please strictly design four diverse and specific question-answer pairs based only on the visible content in the image.

Your goal is to generate VQA data that focuses on the most crucial information for autonomous driving decision-making.

Do NOT create any questions or answers about objects, events, or details that are NOT present in the given image.

The questions should cover, as much as possible, the following critical aspects relevant to driving:

- Lane lines and road boundaries
- Presence, type, and status of traffic signs and lights
- Positions and number of vehicles, pedestrians, or cyclists
- Obstacles or unusual road conditions
- Road directions or upcoming maneuvers

For each image:

- Design 4 questions, each addressing a different key element visible in the scene.
- Provide an accurate and unambiguous answer to each question, strictly based on the image content.
- Do not invent information. Only output what can be directly observed.

Present your output as follows:
```
Q1:  <question 1>
A1:  <answer 1>
Q2:  <question 2>
A2:  <answer 2>
Q3:  <question 3>
A3:  <answer 3>
Q4:  <question 4>
A4:  <answer 4>
```
Please begin when I input the first image.

---

### A.3.2 CONSTRUCTION PIPELINE OF DRIVE-INTERNAL DATASET

The Drive-Internal dataset was curated from a massive database of real-world driving logs through a rigorous pipeline designed to capture high-value, complex driving situations. The construction process consists of three key stages:

1. **Long-tail Instance Retrieval.** We formulated a fine-grained taxonomy to address long-tail distributions, targeting corner cases such as structurally abnormal vehicles, debris, and wildlife. We utilized a CLIP to perform open-vocabulary retrieval from the raw logs. Candidates were subsequently refined through human inspection to ensure high relevance to the target long-tail categories.

2. **Critical Scenario Extraction.** Complementing object-centric mining, we extracted challenging scenarios characterized by dynamic environmental interactions. By computing the variance in driving maneuvers, we isolated clips where the ego-vehicle undergoes significant state changes, thereby capturing moments that test the robustness of planning algorithms.

3. **Decision-Centric Frame Sampling.** To support planning tasks, keyframe selection is tailored to the scenario type. For maneuvers involving speed or directional changes, we select frames 0.5s–1.0s prior to the event, ensuring the model receives input within a realistic reaction threshold. In steady-state driving scenarios, a frame maximizing contextual information is chosen as the representative sample.

A.4   DM-SFT COT DATA CONSTRUCTION PIPELINE

To provide high-quality data for the initial SFT phase, we developed an automated data construction pipeline structured as a three-stage workflow (see Fig. 5). This pipeline systematically processes raw datasets—which exclusively contain multimodal inputs and ground-truth meta-actions—and converts them into comprehensive reasoning datasets that effectively integrate both $\mathcal{M}_{\text{tool}}$ and $\mathcal{M}_{\text{text}}$ reasoning modalities.

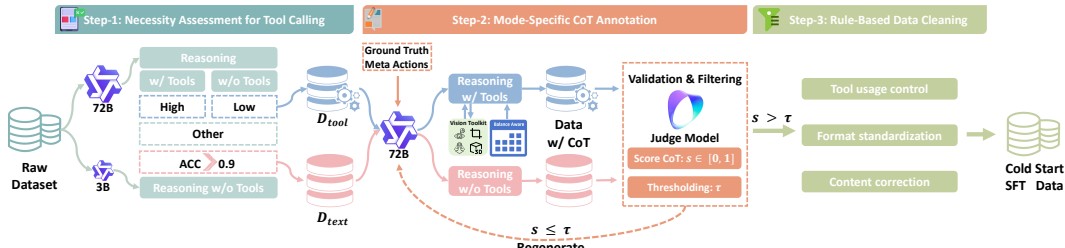

Figure 5: The automated three-stage pipeline for SFT data construction. The process consists of: (1) Necessity Assessment, which partitions data into tool-necessary $D_{\text{tool}}$ and tool-unnecessary $D_{\text{text}}$ sets; (2) Mode-Specific CoT Annotation, where a teacher model's outputs are refined through a validation-regeneration loop with a judge model; and (3) Rule-Based Data Cleaning to produce the final dataset.

**Step 1: Necessity Assessment for Tool Calling.**   This initial step partitions the raw dataset by discerning the necessity of tool use for each sample. Our two-model approach aims to establish clear boundaries for scenario complexity. We first employ a lightweight Qwen2.5-VL-3B model to filter out simple scenarios where high accuracy (e.g., $> 0.9$) is achievable in the $\mathcal{M}_{\text{text}}$ mode, forming the tool-unnecessary set $D_{\text{text}}$. For the remaining samples, a powerful Qwen2.5-VL-72B oracle model determines the tool-necessary set $D_{\text{tool}}$ by identifying cases where the $\mathcal{M}_{\text{tool}}$ mode yields a significant performance gain over the $\mathcal{M}_{\text{text}}$ mode. This use of a highly capable oracle provides a reliable, conservative criterion for tool necessity, ensuring that we identify scenarios where external perception is genuinely beneficial, while leaving ambiguous cases for the exploratory set $D_{\text{explore}}$ used in subsequent RL stages.

**Step 2: Mode-Specific CoT Annotation and Validation.**   In this step, we generate mode-specific CoT annotations using the powerful Qwen2.5-VL-72B as the "teacher model." For each classified sample, the teacher model performs "inverse" reasoning from the ground-truth meta-actions to construct a logical thought process in either the $\mathcal{M}_{\text{text}}$ or $\mathcal{M}_{\text{tool}}$ mode. By anchoring the teacher model to the ground-truth action, we effectively steer the reasoning process. This approach corroborates recent observations (Zhou et al., 2025) regarding the utility of constrained or prompted generation in producing high-quality annotations. To ensure the quality and coherence of these annotations, we implement a rigorous validation and filtering process. A separate "judge model" scores the logical consistency of each generated CoT on a scale $s \in [0, 1]$. Samples with scores falling below a predefined quality threshold $\tau$ are automatically sent back to the teacher model for regeneration. This iterative refinement loop ensures that only high-fidelity reasoning paths are retained. To ensure a balanced distribution of tool use within the $D_{\text{tool}}$ set, a **Dynamic Tool Preference Guidance mechanism** is also employed during this process, preventing a preference tool bias in the annotated data.

**Step 3: Rule-Based Data Cleaning.**   In the final step, the annotated data is subjected to a rule-based cleaning pipeline to ensure its fidelity and uniformity. This pipeline performs three primary functions: (1) **Format Standardization**, which validates the structural integrity of the CoT and the syntactical correctness of tool calls; (2) **Content Correction**, which rectifies content-level errors such as annotation artifacts or mismatches between the final prediction and the ground-truth label; and (3) **Tool Usage Control**, which normalizes tool usage by constraining call frequency and eliminating redundant or illogical calls.

After this pipeline, we sample an equal number of high-quality instances from $D_{\text{text}}$ and $D_{\text{tool}}$ to construct the final cold-start SFT dataset, providing a solid foundation for the model's hybrid-thinking capabilities.

## A.5   REWARD DESIGN DETAILS

**Accuracy Reward ($R_{\text{acc}}$).**   The accuracy reward, $R_{\text{acc}}$, is designed to evaluate the quality of the generated meta-action sequence by comparing it to the ground-truth sequence. It is a composite score that combines the accuracy of speed and trajectory predictions, with a higher weight on speed to emphasize safety. The formulation is as follows:

$$R_{\text{acc}} = \lambda_s \cdot R_{\text{speed}} + \lambda_j \cdot R_{\text{traj}}$$

where $\lambda_s = 0.7$ and $\lambda_j = 0.3$ are the weighting factors for the speed reward ($R_{\text{speed}}$) and trajectory reward ($R_{\text{traj}}$), respectively.

Both $R_{\text{speed}}$ and $R_{\text{traj}}$ are calculated based on a weighted Levenshtein distance, which measures the similarity between the predicted sequence $\hat{A} = (\hat{a}_1, \ldots, \hat{a}_m)$ and the ground-truth sequence $A = (a_1, \ldots, a_n)$. The similarity score for a sequence is defined as:

$$R_{\text{seq}}(\hat{A}, A) = 1 - \frac{D(\hat{A}, A)}{\max(m, n)}$$

where $D(\hat{A}, A)$ is the weighted Levenshtein distance. The distance is computed recursively. Let $D(i, j)$ be the distance between the first $i$ elements of $\hat{A}$ and the first $j$ elements of $A$. Then:

$$D(i, j) = \min \begin{cases} D(i - 1, j) + C_{\text{del}} \\ D(i, j - 1) + C_{\text{ins}} \\ D(i - 1, j - 1) + C_{\text{sub}}(\hat{a}_i, a_j, j) \end{cases}$$

The costs for deletion ($C_{\text{del}}$) and insertion ($C_{\text{ins}}$) are set to a constant value of 0.6 to allow for more flexible alignment between sequences of potentially different lengths. The substitution cost, $C_{\text{sub}}$, is defined as $1 - S(\hat{a}_i, a_j, j)$, where $S$ is a position-aware match score:

$$S(\hat{a}, a, t) = \mathbb{I}(\hat{a} = a) \cdot w_{a,t}$$

Here, $\mathbb{I}(\cdot)$ is the indicator function, which is 1 if the predicted action $\hat{a}$ matches the ground-truth action $a$, and 0 otherwise. $w_{a,t}$ is the **position-action weight** for the ground-truth action $a$ at timestep $t$.

This weighting mechanism is designed to counteract the data imbalance. By up-weighting rarer, more critical actions (e.g., "Stop", "Left Turn"), we encourage the model to learn a more balanced and robust decision-making policy. The weights are derived from the frequency $f_{c,t}$ of each action class $c$ at each timestep $t$ in the training data. The calculation proceeds in three steps:

1. **Raw Inverse Frequency Weight:** A raw weight $w'_{c,t}$ is calculated based on the inverse frequency relative to the mean frequency $\bar{f}_t$ at that timestep, controlled by an exponent $\gamma = 0.5$:

$$w'_{c,t} = \left( \frac{\bar{f}_t + \epsilon}{f_{c,t} + \epsilon} \right)^{\gamma}$$

2. **Clipping:** The raw weights are clipped to a predefined range $[w_{\min}, w_{\max}]$ (e.g., $[0.7, 1.3]$) to prevent extreme values from destabilizing training:

$$w''_{c,t} = \text{clip}(w'_{c,t}, w_{\min}, w_{\max})$$

3. **Normalization:** Finally, the clipped weights for each timestep are normalized to have a mean of 1, ensuring that the weighting scheme adjusts the relative importance of actions without altering the overall scale of the reward at each timestep:

$$w_{c,t} = \frac{w''_{c,t}}{\frac{1}{|V|} \sum_{c' \in V} w''_{c',t}}$$

where $V$ is the set of all possible actions for the given component (speed or trajectory).

This comprehensive reward design ensures that the agent is rewarded not only for correctness but also for learning to handle critical, less-frequent driving maneuvers effectively.

**Tool Usage Reward** ($R_{\textbf{tool}}$).    In the final AMS-RL stage, the reward function is augmented with a tool-usage component, $R_{\text{tool}}$, to guide the agent in learning *when* to use tools. This reward is only applied to trajectories generated in the tool-use mode ($\mathcal{M}_{\text{tool}}$) and is designed to penalize superfluous tool calls while rewarding effective ones. The core mechanism is a **contrastive evaluation** within each group of $G$ generated responses for a given input.

First, we establish a performance baseline, $\bar{R}_{\text{acc}}^{\text{text}}$, by calculating the average accuracy of all text-only trajectories within the group. **If a group contains no such text-only paths, this baseline cannot be established, and the $R_{\textbf{tool}}$ reward is not applied (i.e., set to zero) for that group, as no contrastive comparison is possible.** For groups where a baseline exists, the reward for each tool-assisted trajectory $i$ is determined by its performance gain over this baseline, adjusted for a penalty corresponding to the number of tool calls.

The tool usage reward, $R_{\text{tool}}$, for a trajectory $i$ that uses $N_i$ tools is calculated as:

$$R_{\text{tool}} = (R_{\text{acc},i} - \bar{R}_{\text{acc}}^{\text{text}}) - N_i \cdot C_{\text{tool}}$$

**Parameter Derivation:** We set the per-call cost coefficient $C_{tool}$ based on a "Minimum Effective Gain" principle rather than empirical tuning. We posit that active perception is justified only if it corrects at least one meta-action in the 4-step sequence, corresponding to an accuracy gain of $0.25$. Amortizing this gain over the median number of expected tool calls (2 calls, given a range of 1-3), we derive $C_{tool} = 0.25/2 = 0.125$. This design explicitly penalizes superfluous tool calls: if the accumulated cost $N_i \cdot C_{tool}$ outweighs the accuracy gain, the agent receives a negative reward, incentivizing it to invoke tools only when they provide a clear performance advantage.

Finally, the reward is clipped to a fixed range, for instance $[-0.2, 0.2]$, to ensure training stability. This design explicitly incentivizes the agent to invoke active perception only when the accuracy improvement is significant enough to overcome the inherent cost of tool usage, fostering an intelligent and efficient decision-making process.

## A.6 TRAINING AND EVALUATION PROMPT TEMPLATE

---

**Prompt Template for Model Training and Evaluation**

<**context**>
You are provided with a front-view image of the car camera from the current frame.
The navigation command is: {navigation_command}.
Your current speed is: {speed} km/h.

<**task**>
Your task is to analyze the driving scenario, construct a natural, logical reasoning process and predict meta-actions. The following three stages are guidelines for structuring your thoughts:
1. <description>: Briefly describe the key aspects of the driving environment relevant to the decision. (<4 sentences)
2. <reasoning>: Briefly explain logically how the perceived situation helps you to predict the meta-actions. (<4 sentences)
3. <prediction>: Based on reasoning, briefly explain how you arrived at the meta-actions. (<2 sentences)

Finally, output the meta-actions for the current frame and the next 3 frames (2s interval between frames).
Output ONE composite action for each frame, each consisting of one speed token from {"Stop", "Keep Speed", "Accelerate", "Decelerate"} and one trajectory token from {"Left Turn", "Right Turn", "Straight"}.
Note that Left Turn and Right Turn include both major turns and minor adjustments like lane changes or small heading corrections.

<**rule**>
• Your response should be concise, prefer short, direct sentences.
• First check if context suffices for prediction, then choose <think_with_tools> or <think_no_tools>.

<**tool_usage**>
Under <think_with_tools>, you need to consider the following content:
• You are recommended to call the tools multiple times for better understanding (max 3 calls).
• Please call the tool meaningfully, as each call to the tool incurs additional costs.

**Example:**

```
<think_with_tools>(or <think_no_tools>)
    <description>
        ...(you can call tools here under <think_with_tools>)
    </description>
    <reasoning>
        ...(you can call tools here under <think_with_tools>)
    </reasoning>
    <prediction>
        ...(you can call tools here under <think_with_tools>)
    </prediction>
</think_with_tools>(or </think_no_tools>)
<meta actions>['Speed, Traj', 'Speed, Traj', 'Speed, Traj', 'Speed,
    Traj']</meta actions>
```

---

## A.7 AUTOMATED META-ACTION LABELING PIPELINE

To generate high-quality training labels, we developed a robust automated pipeline that converts raw vehicle trajectory data into discrete meta-action sequences. Distinguishing itself from static rule-based or direct VLM generation approaches, our method employs an Iterative Dual-Feedback

Optimization strategy. This pipeline optimizes kinematic rule thresholds through a cycle of VLM-based consistency checks and human verification, ensuring that the final labels are both physically plausible and semantically accurate before undergoing final refinement for complex cases.

### A.7.1   RULE-BASED GENERATION

The core of our pipeline is a rule-based labeling process that maps continuous trajectory data to a predefined vocabulary of speed and trajectory actions. This process is governed by a sliding window mechanism.

**Sliding Window Mechanism.**   We employ a sliding window that spans a 3-second duration, encompassing 1 second of historical trajectory and 2 seconds of future trajectory relative to its center point. Starting from the current frame, the window slides forward four times in 2-second increments. This procedure generates a four-step discrete meta-action sequence, representing the planned behavior for the next 8 seconds at a 2-second resolution.

**Trajectory-to-Action Mapping.**   For each segment of the trajectory captured by the window, we derive a composite meta-action consisting of a longitudinal (speed) and a lateral (trajectory) component based on the following rules:

- **Longitudinal Action (Speed):** The speed action is determined by the vehicle's average speed and acceleration within the window.
  - **Stop:** If the speed is below a threshold of $0.5\,m/s$.
  - **Accelerate / Decelerate:** If the absolute acceleration exceeds a threshold of $0.3\,m/s^2$.
  - **Keep Speed:** If the absolute acceleration is within the $\pm 0.3\,m/s^2$ range.
- **Lateral Action (Trajectory):** The trajectory action is classified by calculating the overall angle between the initial and final direction vectors of the trajectory segment within the window.
  - **Straight:** If the absolute angle of deviation is less than $15°$.
  - **Left Turn / Right Turn:** If the angle exceeds $15°$, with the direction determined by the sign of the angle.

**Physically Grounded Thresholds.** The thresholds employed in our rules are physically grounded to ensure interpretability. Specifically, the speed threshold of $0.5\ m/s$ effectively distinguishes intentional movement from sensor noise or station-keeping drift. Similarly, the acceleration threshold of $0.3\ m/s^2$ captures deliberate velocity changes while filtering out minor fluctuations typical of cruising. The heading deviation of $15°$ reliably separates straight driving from intentional lateral maneuvers. Crucially, these values were not arbitrarily chosen heuristics; rather, they are the converged results of an iterative "Dual-Feedback" optimization process, which effectively bridges kinematic rules with semantic understanding, as detailed in the following section.

### A.7.2   ITERATIVE THRESHOLD OPTIMIZATION AND VLM-BASED REFINEMENT

While rule-based mapping provides a consistent baseline, determining the optimal boundaries for "intentional" driving actions can be ambiguous. To ensure high-quality labels and address the known limitations of VLMs in temporal grounding, we implemented a "Rule-Based Generation + VLM-as-Judge + Human Verification" pipeline with a dual-feedback mechanism.

**VLM-as-Judge with BEV Visualization.** In this step, we leverage the advanced reasoning capabilities of GPT-4.1 to act as a proxy for human verification. We provide the model with a bird's-eye-view (BEV) visualization of the ego-vehicle's future trajectory alongside the initial rule-based meta-action sequence. This strategy of providing explicit visual context effectively mitigates the temporal grounding weaknesses often observed in VLMs, a challenge also highlighted and similarly addressed in Xu et al. (2025b) through visual prompts. The model is then tasked with evaluating the sequence for logical consistency.

**Dual-Feedback Iterative Tuning.** We treated the kinematic thresholds (speed, acceleration, yaw) as parameters within a dual-feedback loop: 1) VLM Feedback: The VLM scored the consistency between the rule-generated labels and the BEV visualizations. 2) Human Feedback: We conducted

random spot-checks to calculate the false-label rate. In each iteration, thresholds were adjusted to simultaneously maximize the VLM's consistency alignment score and minimize the human-observed error rate. The thresholds reported in the previous section represent the converged values from this optimization.

**Final Refinement for Hard Cases.** For the small fraction of remaining boundary cases where rule-based outputs remained ambiguous (low VLM confidence), we performed a final pass using GPT-4.1. By providing the model with the BEV visualization and the tentative rule-based sequence, we allowed it to refine the labels based on logical consistency and scene context. The prompt used for this final refinement is detailed below.

---

**VLM Refinement Prompt**

**System Prompt:**
You are an expert driving meta-action evaluator and corrector. You will be given a bird's-eye-view (BEV) image where a blue polyline indicates the future ego trajectory. You will also receive a 4-step rule-based meta-action sequence, each step being a pair: [speed_action, trajectory_action]. Your job: (1) rate how consistent the sequence is with the visible trajectory shape and plausible driving behavior, (2) correct any unreasonable actions.

---

**User Prompt:**
Consider the image and the rule-based 4-step meta-action sequence.

- **Allowed speed actions:** Stop, Keep Speed, Accelerate, Decelerate.
- **Allowed trajectory actions:** Left Turn, Right Turn, Straight.
- Return a strict JSON object with keys: score (0-10 integer), reason, and final_meta_actions.
- final_meta_actions MUST be a list of EXACTLY 4 strings, preserving the original speed action for each step.
- Each string MUST be in the form '<speed>, <trajectory>' using ONLY the allowed labels.
- **Temporal semantics:** the 4 steps cover [0-2s], [2-4s], [4-6s], [6-8s] after the current frame; the first step is the current 0-2s window.
- **Trajectory evidence:** ONLY use the first 16 future trajectory points starting from the current ego position; ignore farther points.
- **Global trend tolerance:** a turning trend does NOT require all steps to be turning; sequences like 'Straight, Straight, Left Turn, Left Turn' are acceptable.
- **Stationary cases:** if future points are nearly stationary (clustered at one location), DO NOT infer a turn; set the trajectory action to 'Straight' for those windows.
- **Plausibility filter:** only fix clearly unreasonable trajectory actions given the geometry and the 2s window (e.g., 'Stop, Left Turn' -> 'Stop, Straight').
- Do not add any commentary. Output only the JSON.

---

### A.8 HUMAN EVALUATION DETAILS

To establish a credible human performance benchmark, we recruited three licensed drivers, each possessing over two years of driving experience. The evaluation tasked these participants with the same high-level planning challenge assigned to the agent: predicting a sequence of four meta-actions for each driving scenario. To facilitate this, we developed a dedicated evaluation interface, as depicted in Figure 6. For each sample, this interface presented the forward-facing camera view, current vehicle speed, and the navigation directive. Participants then selected the appropriate meta-actions for the subsequent 8-second horizon using a series of dropdown menus. The test sets from both Drive-Interal and nuScenes were partitioned equally, with each participant evaluating a unique third of the samples. The final "Human" baseline score reported in the main results is the average of the accuracies from these three evaluators.

**Human Evaluation**

Figure 6: The user interface for human evaluation. For each scenario, evaluators were provided with the front-view camera image, a navigation instruction, and the current vehicle speed. They used the dropdown menus to predict the 4-step meta-action sequence for the upcoming 8-second interval.

## A.9    EXTENDED EXPERIMENTAL RESULTS

### A.9.1    DETAILED RESULTS FOR DOMAIN ALIGNMENT

To provide a comprehensive view of the benefits of our autonomous driving domain alignment strategy, this section presents the detailed performance breakdown of our *DriveAlign-3B* model. Table 9 compares its performance against the base model, Qwen2.5-VL-3B, and the larger Qwen2.5-VL-7B across our domain-specific DriveAlign-VQA$_{test}$ benchmark and a suite of eight general VLM benchmarks.

As referenced in the main paper, the results demonstrate that *DriveAlign-3B* not only achieves a significant improvement of +11.7 points in the domain-specific overall score but also enhances its general VLM capabilities. This dual improvement validates our approach of developing robust driving-specific priors without compromising, and in many cases strengthening, the model's foundational reasoning abilities.

Table 9: **Detailed results of domain alignment on in-domain and general abilities.** Scores on DriveAlign-VQA$_{test}$ (upper) and 8 general VLM benchmarks (lower). Numbers in blue/red are deltas vs. Qwen2.5-VL-3B.

| Task | Benchmark | DriveAlign-3B (ours) | Qwen2.5-VL-3B | Qwen2.5-VL-7B |
|---|---|---|---|---|
| DriveAlign-VQA$_{test}$ | scene description | 87.8 (+5.5) | 82.3 | 82.2 |
| | recognition of traffic entities | 91.5 (+11.4) | 80.1 | 86.4 |
| | localization of critical targets | 85.1 (+12.9) | 72.2 | 67.9 |
| | traffic commonsense and rules | 74.0 (+17.1) | 56.9 | 71.5 |
| | **overall** | **84.6** (+11.7) | **72.9** | **77.0** |
| General | MMStar | 56.7 (+2.9) | 53.8 | 60.6 |
| | MMVet | 60.6 (-0.6) | 61.2 | 69.0 |
| | MMBench_EN (val) | 80.4 (+1.1) | 79.3 | 79.7 |
| | MMBench_EN_V11 (val) | 77.4 (+1.0) | 76.4 | 80.3 |
| | MME-Realworld-lite | 48.3 (+6.2) | 42.1 | 44.6 |
| | OCRBench | 81.4 (-1.2) | 82.6 | 88.1 |
| | MMMU (val) | 46.0 (+4.4) | 41.6 | 48.4 |
| | AI2D | 80.1 (+1.9) | 78.2 | 80.7 |
| | **overall** | **66.4**(+2.0) | 64.4 | 68.9 |

**Notes.** For DriveAlign-VQA$_{test}$, each of the four topics contains 200 samples; the judge model is Doubao Seed 1.6. The general benchmarks suite includes MMStar (Chen et al., 2024), MMVet (Yu et al., 2024), MMBench (Liu et al., 2024a), MME-Realworld-lite (Zhang et al., 2024), OCRBench (Liu et al., 2024b), MMMU (Yue et al., 2024), and AI2D (Kembhavi et al., 2016). The general evaluation was performed using VLMEvalKit (Duan et al., 2024) with the default configuration.

### A.9.2    EXTENDED ABLATION RESULTS FOR PROGRESSIVE TRAINING STRATEGY

To further validate the robustness of our progressive training strategy, we present the ablation study results on the nuScenes dataset in Table 10. These results complement the main analysis conducted on

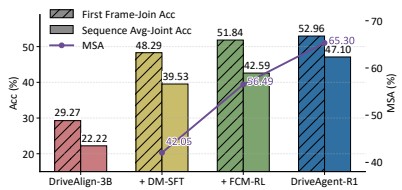

Figure 7: **Progressive training gains on nuScenes test set.** Accuracy in the adaptive mode ($\mathcal{M}_{\text{adaptive}}$) and MSA improve with each training stage.

Table 10: **Ablation on the progressive training strategy on the nuScenes test set.** We evaluate different combinations of our three training stages. To ensure fair comparison, we also test variants where a single RL stage (FCM-RL or AMS-RL) is trained for two epochs, matching the total RL epochs of our final model. Best results are in **bold**.

| Variant | Training Stage | | | Sequence Avg-Joint Acc (%) | | | MSA (%) |
|---|---|---|---|---|---|---|---|
| | DM-SFT | FCM-RL | AMS-RL | $\mathcal{M}_{\text{tool}}$ | $\mathcal{M}_{\text{text}}$ | $\mathcal{M}_{\text{adaptive}}$ | |
| Variant-1 (SFT Only) | 1 epoch | - | - | 41.04 | 39.17 | 39.53 | 42.05 |
| Variant-2 (+FCM) | 1 epoch | 1 epoch | - | 42.92 | 40.58 | 42.59 | 56.49 |
| Variant-3 (+FCM x2) | 1 epoch | 2 epochs | - | 45.15 | 41.07 | 44.60 | 58.30 |
| Variant-4 (+AMS) | 1 epoch | - | 1 epoch | 41.18 | 40.58 | 40.92 | 57.76 |
| Variant-5 (+AMS x2) | 1 epoch | - | 2 epochs | 44.19 | 41.47 | 43.57 | 60.30 |
| **DriveAgent-R1 (FCM→AMS)** | **1 epoch** | **1 epoch** | **1 epoch** | **46.38** | **44.43** | **47.10** | **65.30** |

the Drive-Internal dataset in Section 4.3.2. The performance trends observed on nuScenes are highly consistent with those on Drive-Internal, confirming that each stage of our training pipeline—from SFT to the cascaded FCM-RL and AMS-RL—contributes to a steady and significant improvement in the agent's capabilities. This cross-dataset consistency underscores the general effectiveness and reliability of our proposed training methodology.

## A.10 QUALITATIVE RESULTS

This section presents qualitative examples to illustrate the capabilities of *DriveAgent-R1* in diverse driving scenarios. We begin with a simple case where the agent efficiently employs text-only reasoning (See Fig. 8 & Fig. 9). Subsequently, we showcase the agent's active perception: at a busy intersection, it uses RoI Inspection and Retrieve View to check signals and pedestrian flow (See Fig. 10); in a low-visibility night scene, it leverages 3D Object Detection to uncover a hidden pedestrian (See Fig. 11); when approaching a barrier, the Depth Estimation tool provides the spatial awareness to navigate safely and plan ahead (See Fig. 12). Finally, two additional examples highlight the crucial role of RoI Inspection in resolving ambiguity by assessing the risk of distant pedestrians and confirming the state of a faraway traffic signal (See Fig. 13). Finally, to explicitly validate the superiority of our *Hybrid Thinking* framework, we present two comparative analyses against rigid "forced-mode" baselines. These cases demonstrate how the adaptive mode outperforms forced strategies by ensuring intent-driven tool usage in complex scenarios (See Fig. 14) and preventing information redundancy in straightforward scenes (See Fig. 15).

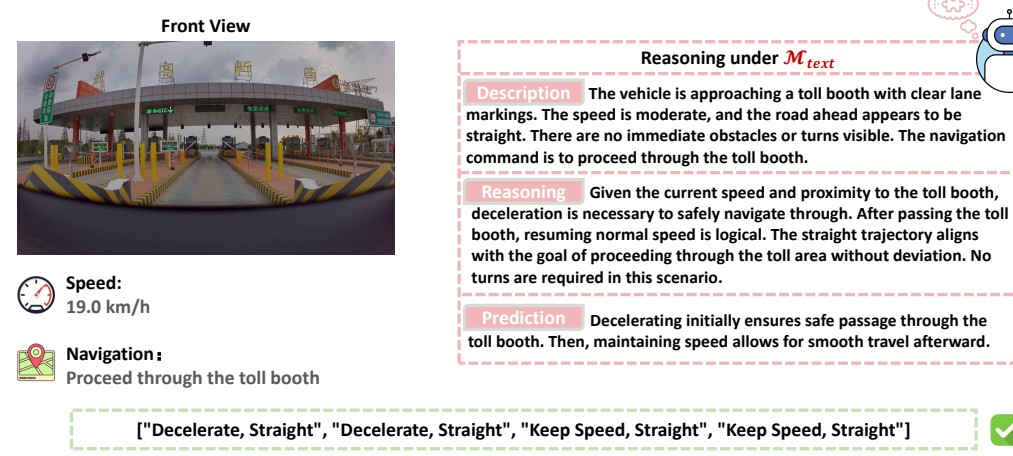

Figure 8: In this routine driving scenario, the vehicle approaches a toll booth with clear lane markings and straightforward navigation. Recognizing the low complexity and lack of ambiguity, *DriveAgent-R1* employs its efficient $\mathcal{M}_{\text{text}}$ mode. Through text-only reasoning, it formulates a safe and logical plan: decelerate to pass through the toll gate, then maintain speed to proceed, effectively handling a common situation without needing to invoke its vision toolkit.

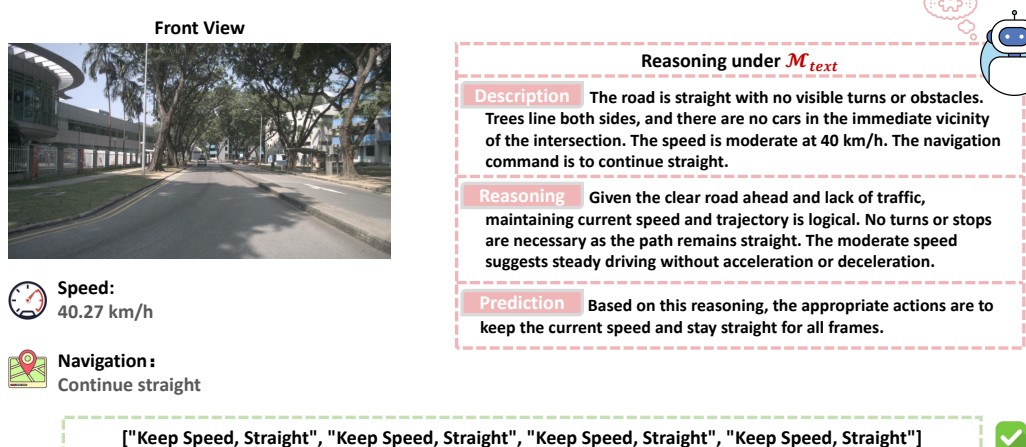

Figure 9: In this case, the road is straight and the view is open with no apparent obstacles, *DriveAgent-R1* engages its efficient text-only reasoning mode ($\mathcal{M}_{\text{text}}$) due to the low scene complexity. By solely analyzing the textual scene description and navigation command, it accurately assesses the simple road conditions and formulates a plan to "Keep Speed, Continue Straight." This decision is made without invoking the vision toolkit, demonstrating the efficiency of the hybrid-thinking framework.

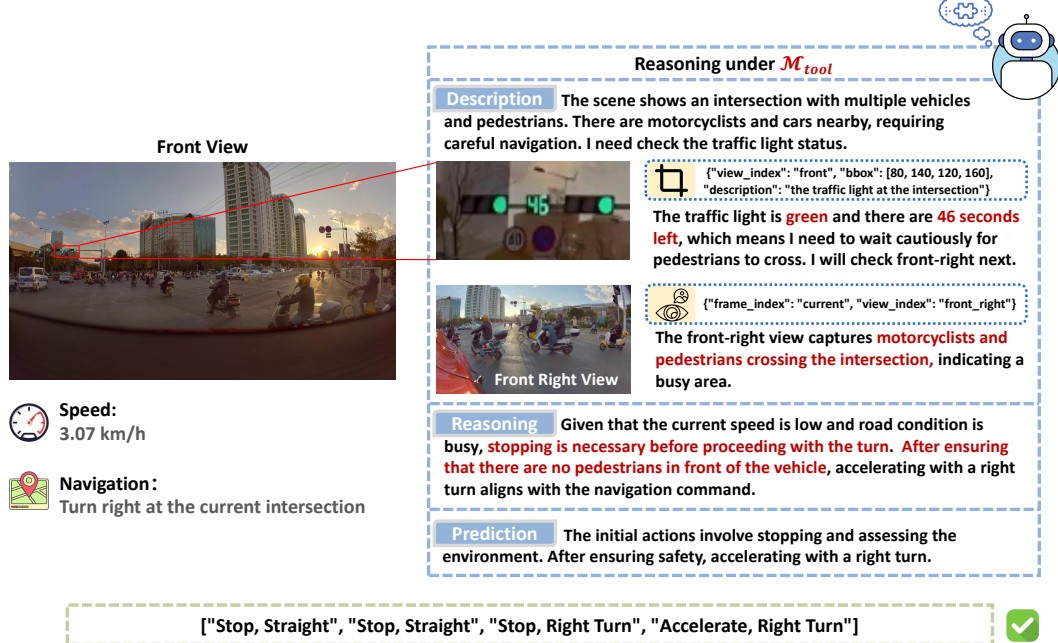

Figure 10: This case highlights the necessity of active perception when navigating a challenging right turn at a busy intersection with heavy pedestrian and motorcycle traffic. The initial view provides insufficient detail to ascertain the traffic light's status, creating uncertainty. To resolve this, *DriveAgent-R1* proactively engages its $\mathcal{M}_{\text{tool}}$ mode. It first invokes the "RoI Inspection" tool to confirm the green light and then use the "Retrieve View" to get the front-right view to better assess the dense flow of crossing traffic. By grounding its reasoning in this actively acquired visual evidence, the agent formulates a safe, context-aware plan: to stop and wait for pedestrians to clear before executing the right turn.

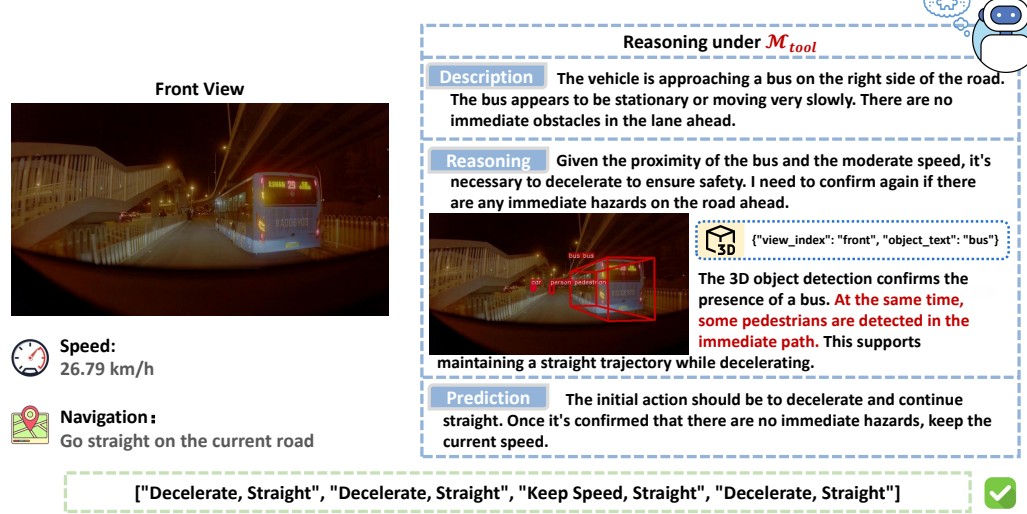

Figure 11: This case demonstrates the agent's ability to uncover potential hazards in challenging low-visibility conditions. While navigating a narrow road at night, the initial perception identifies a nearby bus but fails to detect a pedestrian in the immediate path due to the darkness. By invoking the "3D Object Detection" tool, *DriveAgent-R1* performs active perception and successfully identifies the otherwise overlooked pedestrian. This critical, tool-grounded discovery prompts the agent to revise its initial assessment and decide to decelerate, prioritizing safety in response to the newly revealed risk.

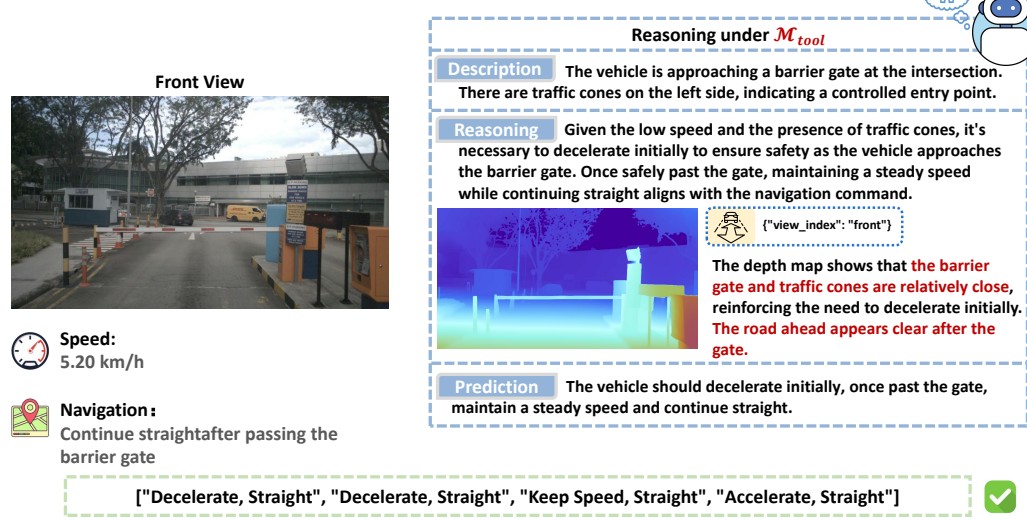

Figure 12: In scenarios requiring precise distance judgment, such as passing through this barrier gate, the "Depth Estimation" tool proves invaluable. The generated depth map provides two critical insights: it first confirms the gate's close proximity, reinforcing the decision to decelerate for safe passage. More importantly, it also perceives the clear and open road beyond the barrier. This foresight into the upcoming environment allows *DriveAgent-R1* to formulate a confident, multi-stage plan: slow down to navigate the immediate obstacle, and then accelerate once safely through.

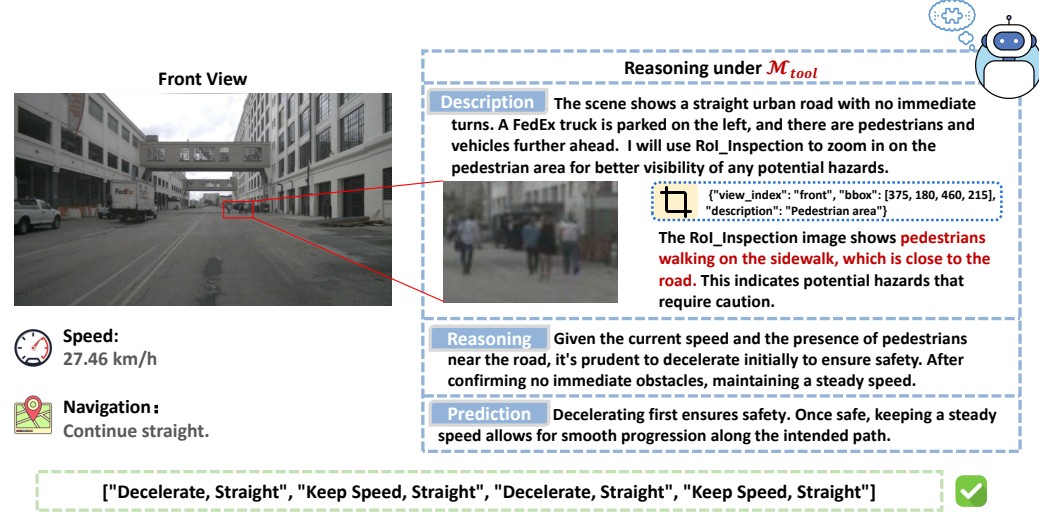

Figure 13: Even on a seemingly clear, straight road, *DriveAgent-R1* exhibits proactive caution by investigating distant activity. While the initial view shows pedestrians far ahead, their exact proximity to the road is uncertain. By deploying the "RoI Inspection" tool, the agent obtains a magnified view, revealing that the individuals are walking very close to the lane of travel. This specific, tool-acquired insight elevates the potential risk, prompting the agent to prudently decelerate to ensure a safe passage.

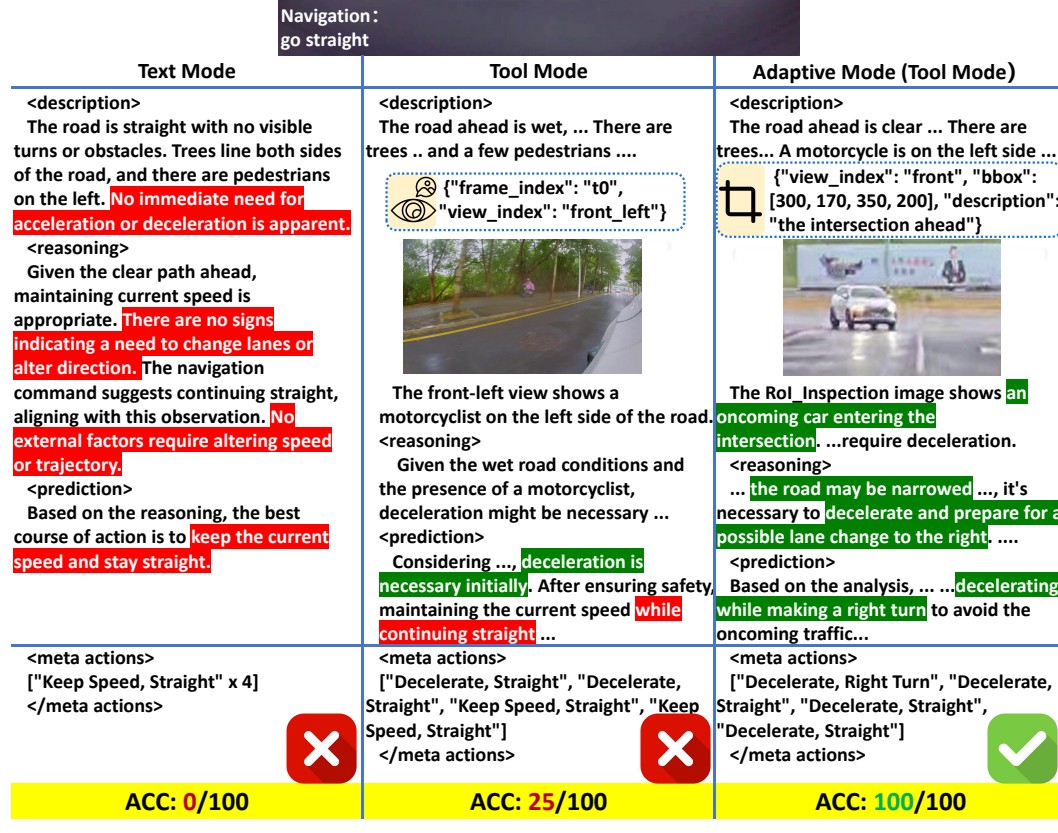

Figure 14: **Comparison of reasoning modes in a complex intersection scenario.** The key object is a distant white car approaching from the intersection (top). Left ($\mathcal{M}_{text}$): Relying solely on the defauplt resolution, the model fails to detect the distant vehicle and incorrectly decides to maintain speed. Center ($\mathcal{M}_{tool}$): Although forced to use tools, the model lacks specific intent. It calls a generic front-left view but fails to focus on the critical area. This illustrates that without the autonomous reasoning to seek specific evidence, tool usage can be ineffective. Right ($\mathcal{M}_{adaptive}$): The model identifies uncertainty and autonomously switches to Tool Mode. It proactively uses RoI Inspection to zoom in on the intersection, successfully identifying the oncoming car and planning a safe lane change. This demonstrates the superiority of active, intent-driven perception over passive tool invocation.

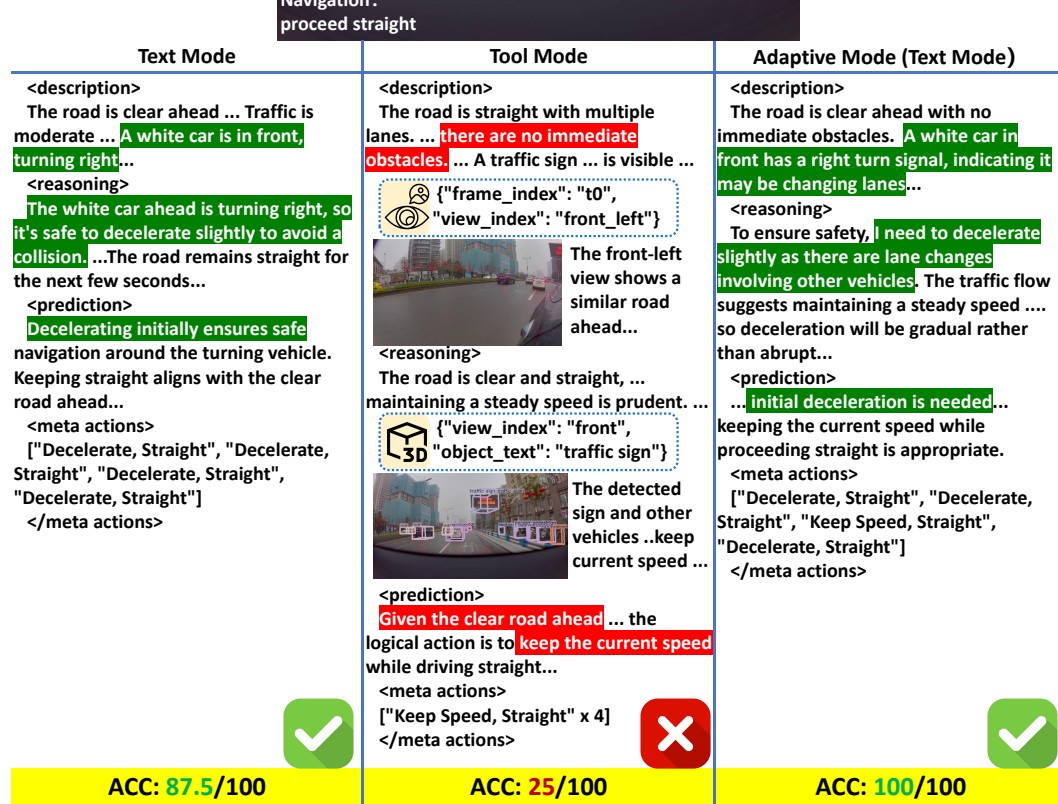

Figure 15: **Comparison of reasoning modes in a clear, high-dynamic scenario.** The key object is a white car cutting into the ego-lane (top). Center ($\mathcal{M}_{tool}$): When forced to use tools in a clear scenario, the model becomes distracted. It invokes 3D Object Detection for a non-critical traffic sign, introducing irrelevant information that diverts attention from the immediate braking event, leading to a dangerous "Keep Speed" prediction. Right ($\mathcal{M}_{adaptive}$): The model correctly recognizes that the initial visual context is sufficient. It selects Text Mode, avoiding the noise and cognitive load of unnecessary tool calls. By focusing on the clear visual evidence of the cut-in, it correctly predicts the necessary deceleration. This highlights the importance of Hybrid Thinking in preventing "tool abuse" and information overload.

A.11   ANALYSIS OF A FAILURE CASE IN COMPLEX INTERSECTIONS

To provide a transparent assessment of our agent's current limitations, this section analyzes a representative failure case in a complex urban intersection (see Figure 16). In this scenario, the agent initially correctly identifies its traffic light as green. However, after proactively retrieving a side-view image to better assess the intersection, it misinterprets a red pedestrian signal as its own, causing it to override its correct initial judgment and erroneously plan to stop.

This failure highlights a key limitation: the agent's over-reliance on new visual evidence from its tools, which stems from an insufficient grasp of complex road topologies and traffic rules. Future work will directly address this challenge through two complementary directions: (1) strengthening the agent's foundational understanding of complex road topologies and traffic rules through targeted domain alignment, and (2) cultivating a more nuanced reasoning process capable of resolving informational conflicts. This will involve training the agent on adversarial scenarios with ambiguous visual cues, teaching it to weigh evidence and resolve contradictions rather than uncritically accepting the most recent perception. By implementing these measures, we aim to develop a more robust agent that not only actively perceives its environment but also critically reasons about what it sees, leading to safer and more reliable decision-making.

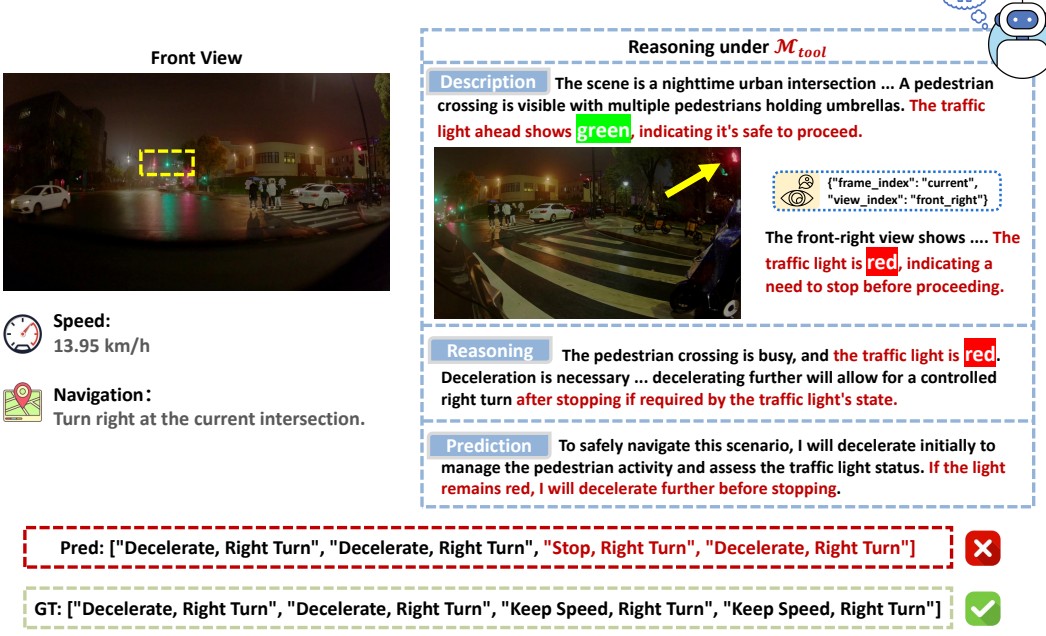

Figure 16: A failure case demonstrating incorrect traffic light identification. The agent initially correctly identifies the forward traffic light as green, but after retrieving a front-right view, it mistakenly focuses on the red pedestrian light, leading to an unnecessary "Stop" command in its final plan.

## A.12 DYNAMIC MODE SWITCHING ANALYSIS

In this section, we include two detailed figure sequences (Fig. 17 and Fig. 19) visualizing the agent's ability to switch between $\mathcal{M}_{text}$ and $\mathcal{M}_{tool}$ based on evolving visual observations.

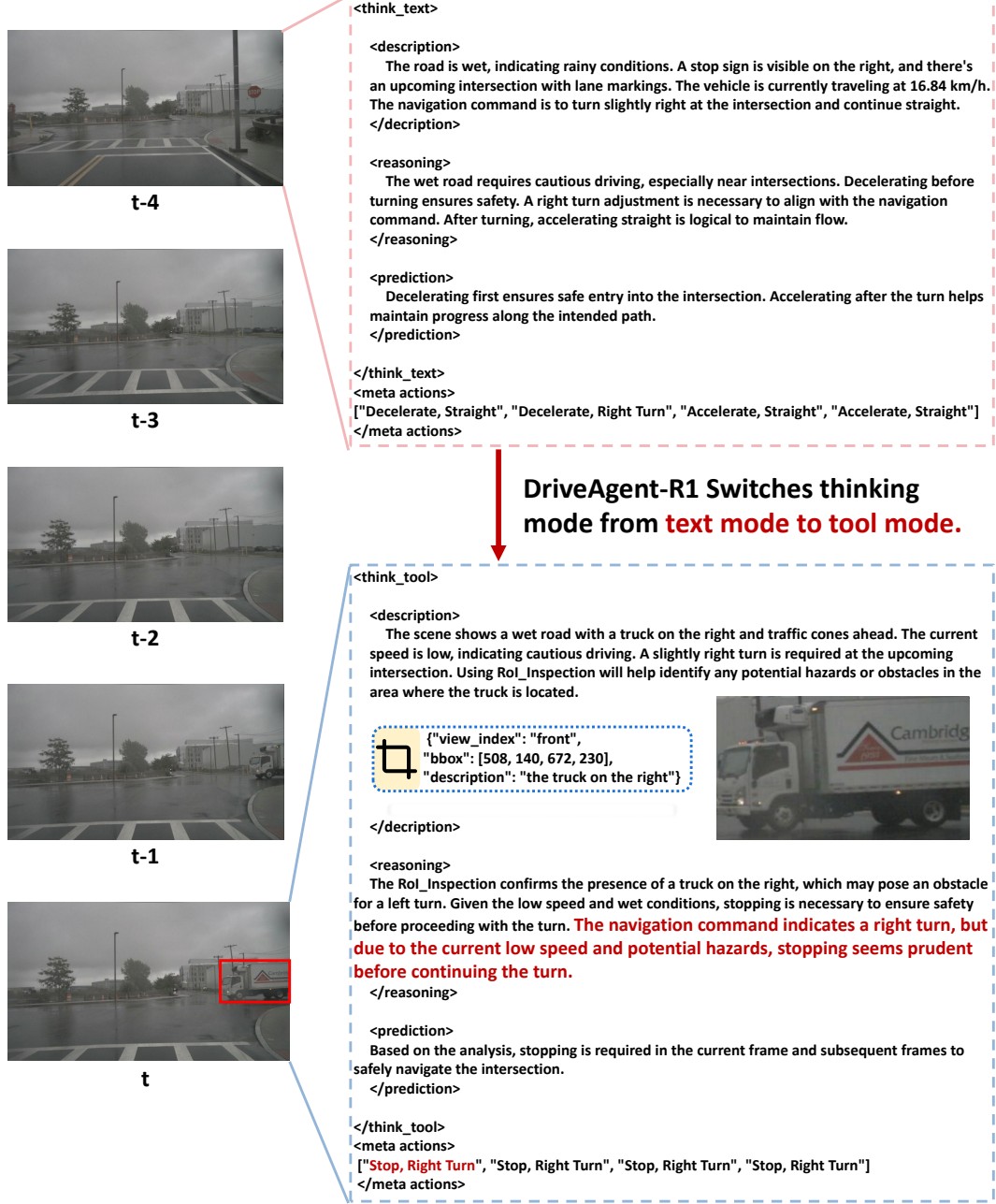

Figure 17: **Dynamic Mode Switching from $\mathcal{M}_{text}$ to $\mathcal{M}_{tool}$ in response to emerging risk.** This sequence illustrates the agent's ability to adapt its reasoning strategy during streaming inference. (Top - $t-4$): The agent operates in Text Mode. Observing the wet road conditions and the upcoming intersection, it relies on internal knowledge to formulate a plan to "Decelerate" and then "Turn Slightly Right." (Bottom - $t$): As the vehicle approaches the intersection, the agent identifies a truck on the right side as a potential conflict. It immediately switches to Tool Mode and invokes RoI Inspection to confirm the target. Grounded by this specific visual evidence, the agent updates its reasoning to prioritize caution, revising its decision to "Stop" to avoid potential collision hazards.

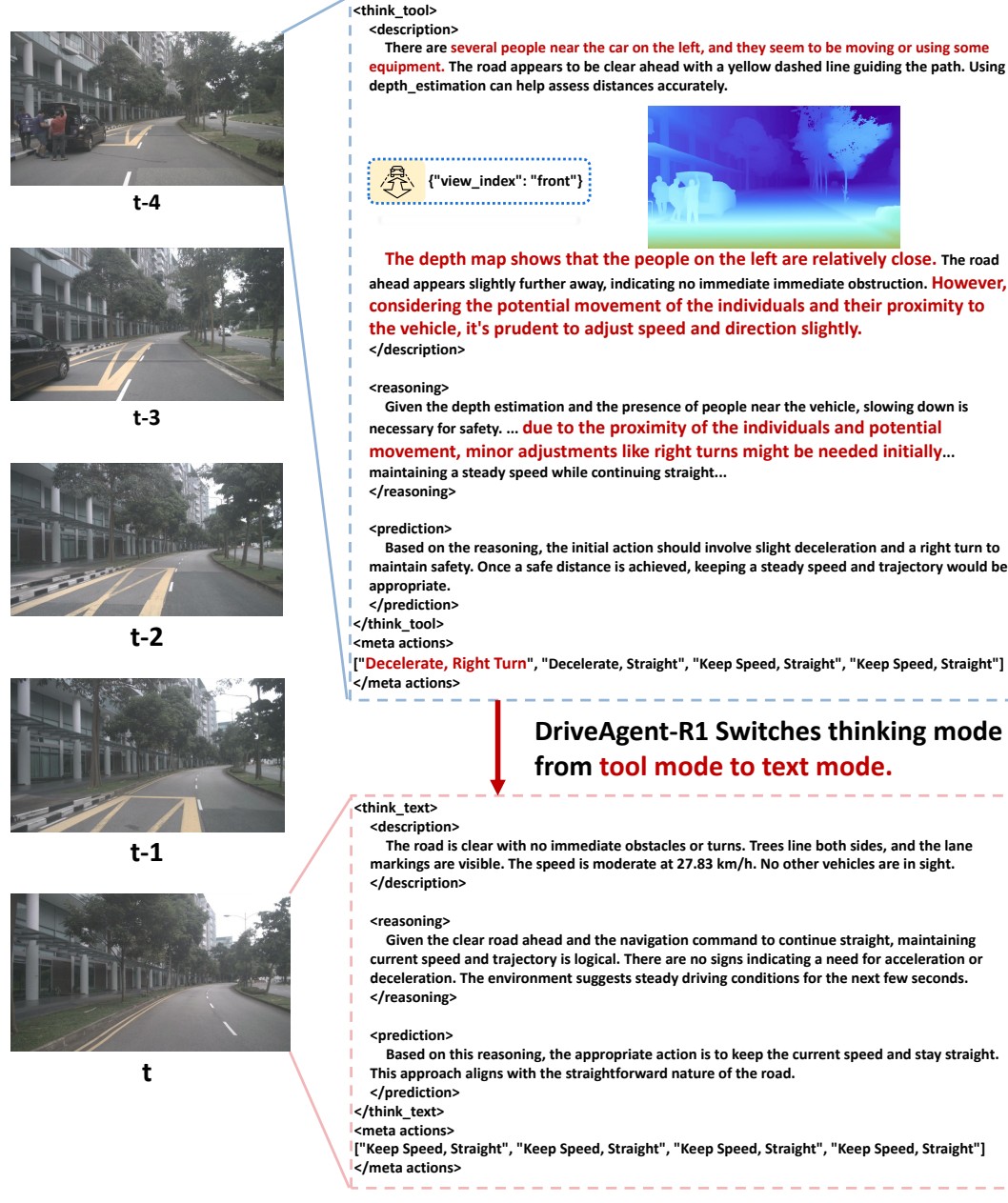

Figure 18: **Dynamic Mode Switching from $\mathcal{M}_{tool}$ to $\mathcal{M}_{text}$ for efficiency.** This sequence demonstrates the agent's capability to optimize inference cost by switching modes based on scene complexity. (Top - $t-4$): The agent encounters pedestrians and equipment on the left. It activates Tool Mode and utilizes Depth Estimation to assess the spatial relationship ("the people on the left are relatively close"). To ensure safety, it plans a maneuver to "Decelerate" and "Turn Right" to increase the lateral distance from the hazard. (Bottom - $t$): After passing the obstruction, the scene simplifies. The agent observes a clear road with no immediate threats and a speed of $27.83$ km/h. It dynamically switches to Text Mode, using efficient internal reasoning to generate a "Keep Speed, Straight" plan, validating that it can autonomously disengage tools when they are unnecessary.

## A.13 COMPARISION WITH INSTRUCTION-GUIDE ATTENTION

In this section, we address the distinction between our tool-based active perception framework and Instruction-Guided Attention (i.e., explicitly prompting the model via text to "focus on" specific regions). While both approaches aim to guide the model's focus, they differ fundamentally in their mechanisms for Information Gain and Generalization.

**Theoretical Distinction:** Resolution Amplifier vs. Attention on Compressed Features Standard VLMs encode input images into a sequence of visual tokens through a fixed-size vision encoder. This process involves downsampling, which inevitably leads to a loss of high-frequency details necessary for resolving small or distant objects (e.g., traffic lights at a distance).

- **Instruction-Guided Attention (Passive)** Providing a text instruction (e.g., "Please pay close attention to the traffic lights in the center") operates strictly within the model's self-attention mechanism. It guides the model to attend to specific visual tokens within the already compressed feature space. Crucially, if the physical visual evidence (e.g., the color of a distant light) was lost during the initial downsampling and encoding, the model cannot "see" it, regardless of how explicit the text instruction is.

- **Active Perception (RoI Inspection)** In contrast, our RoI Inspection tool acts as a resolution amplifier. It accesses the lossless, raw image buffer. When the agent predicts a bounding box, the tool extracts this region from the high-resolution source and re-encodes it. This process introduces new, high-fidelity pixel information that was absent in the global view tokens, effectively bypassing the resolution bottleneck of the base encoder.

### A.13.1 QUANTITATIVE ANALYSIS

To empirically validate this distinction, we conducted a comparative study on the Drive-Internal test set. We compared DriveAgent-R1 (equipped with active tools) against a baseline using Textual Attention Prompting. Since manually tailoring specific coordinates for every test case is infeasible and would require an oracle, we utilized a generic, scene-agnostic instruction designed to cover common critical objects: "Please pay close attention to the vehicles and traffic lights in the center of the image."

Table 11: Comparison of Active Perception vs. Instruction-Guided Attention on Drive-Internal test set. **Bold** denotes the best performance.

| Inference Setting | First-Frame Joint Acc. | Seq. Avg. Joint Acc. |
|---|---|---|
| Baseline ($\mathcal{M}_{text}$) | 45.27 | 43.29 |
| Instruction-Guided Attention | 46.72 | 43.83 |
| **Active Perception (RoI)** | **48.81** | **44.65** |

The results in Table 11 demonstrate that while guiding the model's attention via text instructions offers a slight improvement, Active RoI Inspection yields a significantly larger gain (+3.54% First-Frame Acc and +1.36% Seq Acc). This confirms that guiding attention within a low-resolution feature map is insufficient for fine-grained planning tasks; increasing the effective resolution is required.

### A.13.2 QUALITATIVE ANALYSIS

In this section, we present a representative failure case of Instruction-Guided Attention.

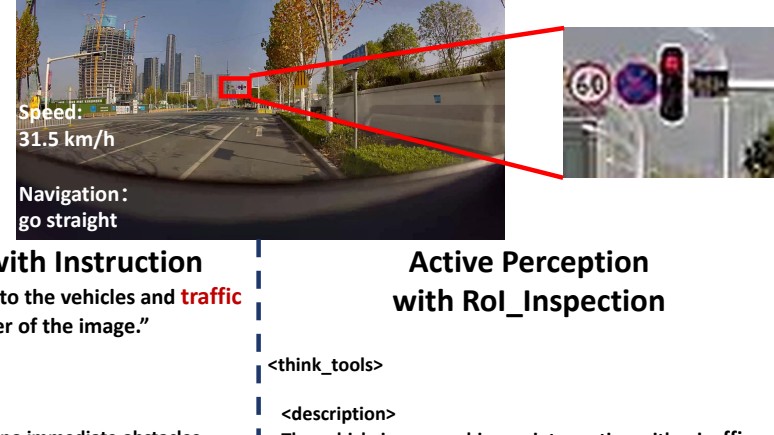

**Prompt model with Instruction**

"Please pay close attention to the vehicles and **traffic lights** in the center of the image."

<think_text>

<description>
The road is clear ahead with no immediate obstacles. **Traffic signals are not visible**, but the presence of an intersection suggests caution. A building under construction is on the left.
</description>
<reasoning>
Given the clear path and moderate speed, deceleration might be necessary initially to ensure safety as we approach any unseen hazards. After confirming the intersection is safe, maintaining current speed while proceeding straight aligns with the navigation command.
</reasoning>
<prediction>
Decelerating first ensures safety before resuming normal speed. This matches the logical sequence of actions for the upcoming frames.
</prediction>

</think_text>

meta actions:
["Decelerate, Straight", "Keep Speed, Straight", "Decelerate, Straight", "Keep Speed, Straight"]

**Active Perception with RoI_Inspection**

<think_tools>

<description>
The vehicle is approaching an intersection with a **traffic light visible ahead**. The road appears clear, and there are no immediate obstacles or pedestrians. **To better understand the traffic light status** and any potential hazards, I will use the RoI_Inspection tool on the traffic light.

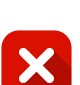 {"view_index": "front", "bbox": [298, 155, 320, 170], "description": "the traffic light in front"}

The RoI_Inspection tool confirms that **the traffic light ahead is red**. This indicates that the vehicle should stop at the intersection.
</description>
<reasoning> Given the red traffic light, ...</reasoning>
<prediction> .... </prediction>

</think_tool>

meta actions:
["Decelerate, Straight", "Decelerate, Straight", "Stop, Straight", "Stop, Straight"]

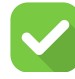

Figure 19: **Comparison of Passive Prompting vs. Active Perception.** Left: Despite being prompted to "pay close attention to the vehicles and traffic lights in the center of the image," the model fails to identify the traffic light status due to the limited resolution of global visual tokens, leading to a risky "Keep Speed" decision. Right: DriveAgent-R1 actively invokes RoI Inspection, retrieving a high-resolution crop that reveals the Red Light, leading to a correct "Stop" decision. This demonstrates that active tools provide critical information gain that static prompts cannot achieve.

