# OpenReview forum: "DriveAgent-R1: Advancing VLM-based Autonomous Driving with Active Perception and Hybrid Thinking"
_ICLR.cc/2026/Conference — ICLR 2026 Poster_

### Official Review · Reviewer_LB8N · 2025-10-23

**Soundness:** 3
**Presentation:** 3
**Contribution:** 2
**Rating:** 4
**Confidence:** 3

**Summary:**

This paper introduces DriveAgent-R1, a vision-language-reasoning framework designed to handle complex, long-tail driving scenarios through hybrid thinking. The core idea is to let the model actively retrieve task-relevant information from tools or external sources to enhance its reasoning about rare or unseen driving cases.

The paper also presents a new QA dataset focused on long-tail driving events, covering diverse edge cases such as unusual pedestrian behavior, obstructed lanes, or ambiguous traffic signals. The authors show that the model’s ability to retrieve external information makes a big difference: without it, performance drops notably (as seen in Table 5).

**Strengths:**

The paper proposes a new hybrid-thinking framework, which combines internal reasoning with external information retrieval. The experiments clearly show that the active retrieval step helps — when it’s removed, performance drops quite a bit (Table 5).

The authors also built a large QA dataset targeting long-tail driving cases. Once released, this dataset should be valuable to the research community, since long-tail reasoning data is still pretty rare for driving tasks.

**Weaknesses:**

The biggest issue here is the choice of baselines. The paper only compares against generalist LLMs like GPT, Gemini, and Qwen. These are not trained for driving at all, so this is really an out-of-domain test for them. DriveAgent-R1, on the other hand, is trained on driving-specific data, so it’s expected to do better. The comparison doesn’t tell us how much progress this paper actually makes within the driving domain.
In DriveVLM, for instance, the authors showed that a domain-specific model can outperform GPT by around 90% — so the marginal improvements here aren’t that surprising. For a fair evaluation, DriveAgent-R1 should be compared against specialized models like DriveLM, DOLPHINS, DriveVLM, or similar. Without that, it’s hard to judge the real impact of this model or dataset.

The evaluation focus is also a bit off. The stated goal is to improve driving and planning, but the experiments don’t report any driving metrics like Displacement Error (DE) or Collision Rate (CR). Other VLM-based systems such as DriveVLM, OpenDriveVLA, and OmniDrive all evaluate directly on these metrics to show actual improvements in control or planning.
Here, DriveAgent-R1 is mostly tested on reasoning or QA accuracy, not on whether it leads to better driving behavior. That leaves a big gap — it’s unclear if this approach really translates into improved driving performance.

**Questions:**

Are there any plans to evaluate DriveAgent-R1 on driving metrics like DE or CR? Without that, it’s hard to tell whether the improvements in reasoning actually help with driving.

Can the authors provide comparisons with domain-specific baselines (e.g., DriveVLM, DriveLM, DOLPHINS)? That would make the results much more meaningful.

---

> ### Author Response · Authors · 2025-11-20
>
> >The biggest issue here is the choice of baselines... For a fair evaluation, DriveAgent-R1 should be compared against specialized models like DriveLM, DOLPHINS, DriveVLM, or similar. Without that, it’s hard to judge the real impact of this model or dataset.
>
> >Can the authors provide comparisons with domain-specific baselines (e.g., DriveVLM, DriveLM, DOLPHINS)? That would make the results much more meaningful.
>
> We sincerely thank you for this constructive feedback. We agree that comparing DriveAgent-R1 against domain-specific models is essential to demonstrate its genuine impact within the autonomous driving domain. Previously, we focused on generalist VLMs due to the significant variance in high-level planning task definitions across different driving-specific works, which made direct comparisons difficult.
>
> To address this concern and provide a fair evaluation, we have conducted new experiments using DriveBench (ICCV 2025), a recently established standardized benchmark for evaluating VLM-based autonomous driving agents on high-level planning task. We evaluated DriveAgent-R1 against DriveLM and DOLPHINS across the benchmark's core metrics.
> | Method | Perception | Prediction | Planning | Behavior |
> | :--- | :---: | :---: | :---: | :---: |
> | DriveLM | 16.85 | **44.33** | **68.71** | 42.78 |
> | Dolphins | 9.59 | 32.66 | 52.91 | 18.81 |
> | **DriveAgent-R1 (Ours)** | **34.07** | 32.85 | 61.89 | **43.69** |
>
> Note: We were unable to include DriveVLM in this comparison as its model weights and code are not open-source.
>
> These results strongly validate the core contributions of our paper:
> 1. **Superior Perception:** DriveAgent-R1 achieves a Perception score of 34.07, doubling the performance of DriveLM (16.85) and tripling that of DOLPHINS (9.59). This quantitative evidence confirms that our Active Perception framework effectively overcomes the limitations of passive perception, allowing the agent to capture critical visual details that other methods miss.
> 2. **Robust High-Level Behavior:** Most importantly, consistent with our paper's core objective of "fine-grained, high-level behavioral planning" (Sec. 1), DriveAgent-R1 achieves the highest score in the Behavior metric (43.69). This demonstrates that our agent successfully translates its superior perceptual grounding into correct high-level driving decisions.
>
> **Changes in Manuscript:** We have added a new subsection, **"3.2.2 Performance on DriveBench," (Page 7)** to the revised manuscript.

---

> ### Author Response · Authors · 2025-11-26
>
> > The evaluation focus is also a bit off. The stated goal is to improve driving and planning, but the experiments don’t report any driving metrics like Displacement Error (DE) or Collision Rate (CR)... it’s unclear if this approach really translates into improved driving performance.
>
> > Are there any plans to evaluate DriveAgent-R1 on driving metrics like DE or CR? Without that, it’s hard to tell whether the improvements in reasoning actually help with driving.
>
> We sincerely apologize for this oversight and thank you for this crucial suggestion. Our original intent was to focus on the efficacy of active perception for high-level intention prediction. However, we agree that demonstrating the translation from high-level thought to low-level execution is essential to validate the system's practical utility.
>
> To address this, we have extended DriveAgent-R1 by integrating a motion planning head to translate high-level decisions into physical trajectories.
> - Architecture: We freeze the pre-trained DriveAgent-R1 (the "Thinker") and attach a lightweight MLP decoder (the "Executor"). The MLP takes three inputs: the predicted meta-action sequence (one-hot encoded), the visual tokens from the Vision Encoder, and the ego-status (velocity and historical trajectory). It predicts the future 3-second trajectory at 2Hz.
> - Training: The MLP head was trained on the nuScenes training set (700 scenes), keeping the VLM backbone frozen.
>
> We evaluated this setup on the nuScenes validation set using standard Open-Loop Planning metrics. The results, compared with state-of-the-art end-to-end methods, are presented below:
>
> | Model | ADE (m) 1s | ADE (m) 2s | ADE (m) 3s | ADE (m) avg | Coll. Rate (%) 1s | Coll. Rate (%) 2s | Coll. Rate (%) 3s | Coll. Rate (%) avg |
> | :--- | :---: | :---: | :---: | :---: | :---: | :---: | :---: | :---: |
> | UniAD | 0.44 | 0.67 | 0.96 | 0.69 | 0.04 | 0.08 | 0.23 | 0.12 |
> | VAD-Base | 0.17 | 0.34 | 0.60 | 0.37 | 0.07 | 0.10 | 0.24 | 0.14 |
> | EMMA | 0.14 | 0.29 | 0.54 | 0.32 | - | - | - | - |
> | DriveVLM-Dual | 0.15 | 0.29 | 0.48 | 0.31 | 0.05 | 0.08 | **0.17** | **0.1** |
> | OmniDrive | 0.14 | 0.29 | 0.55 | 0.33 | **0.00** | 0.13 | 0.78 | 0.3 |
> | OpenDriveVLA (3B) | 0.14 | 0.30 | 0.55 | 0.33 | 0.02 | **0.07** | 0.22 | **0.1** |
> | **DriveAgent-R1 (Ours)** | **0.13** | **0.24** | **0.47** | **0.28** | 0.01 | 0.12 | 0.30 | 0.14 |
> | *w/o meta-actions* | 0.20 | 0.35 | 0.59 | 0.38 | 0.07 | 0.22 | 0.42 | 0.24 |
>
> DriveAgent-R1 achieves the lowest ADE (0.28m) and a comparable Collision Rate against top-tier methods like UniAD and DriveVLM.
>
> **Effectiveness of High-Level Planning:** To isolate the impact of our VLM's High Level Behavior Planning, we conducted an ablation study removing the meta-action input from the MLP ("w/o meta-actions"). Performance dropped significantly (ADE rose from 0.28m to 0.38m; Collision Rate nearly doubled). This confirms that the high-level plans generated by DriveAgent-R1 provide critical guidance for safe and accurate low-level control.
>
> **Changes in Manuscript:** We have added a new **Section 3.3: Performance of Motion Planning (Page 8)** to the main paper, detailing the motion planning head implementation, the comparison with baselines, and the ablation analysis demonstrating the impact of high-level planning on trajectory generation.

---

> ### Author Response · Authors · 2025-11-26
>
> Dear Reviewer LB8N,
> ﻿
>
> We wanted to follow up to see if our response has addressed your concerns. As the discussion period is ending soon, please let us know if you have any remaining questions. If our response has resolved your concerns, we would appreciate it if you could reconsider your score.
>
> Best regards,
>
> The Authors

---

### Official Review · Reviewer_YkZV · 2025-10-30

**Soundness:** 3
**Presentation:** 3
**Contribution:** 3
**Rating:** 6
**Confidence:** 3

**Summary:**

The paper introduces DriveAgent-R1, an autonomous driving agent that incorporates active perception and a hybrid-thinking framework for high-level driving planning. In contrast to previous Vision-Language Models (VLMs), which are limited by passive perception, DriveAgent-R1 dynamically interacts with visual tools to resolve uncertainties in complex driving scenarios. DriveAgent-R1 achieves competitive performance with top models like GPT-5, human drivers, and other VLM-based systems, using only 3B parameters while maintaining deployment efficiency. It outperforms models using passive perception in scenarios requiring active visual inspection.

**Strengths:**

1. The paper introduces active perception to autonomous driving planning, an important advancement beyond the passive perception typically used in VLM-based systems. The integration of tool-based M-CoT for real-time visual reasoning represents a novel approach in this field.
2. The methodology and results are presented in a clear, structured manner, with helpful diagrams and an effective explanation of the agent's reasoning process. The use of visual tools (such as the Vision Toolkit) for active perception is well-described and easy to follow.
3. The hybrid-thinking framework is well-designed, allowing the agent to switch between text-based reasoning and tool-augmented reasoning. The progressive training strategy is a strong contribution, effectively training the agent to adapt its thinking mode based on scene complexity.

**Weaknesses:**

1. There is insufficient statistical validation such as confidence intervals or multiple trials. The reported improvements (e.g., 6.07% gain) would be more convincing with more rigorous statistical analysis.
2. While the Cascaded RL strategy is an interesting approach, the reward design and training stages could be more clearly explained, especially for readers unfamiliar with GRPO. Additionally, the computational cost of such a training strategy could be a concern for real-world deployment.

**Questions:**

1. The performance drops significantly without access to visual tools. How would DriveAgent-R1 handle sensor failures, such as loss of camera feed or occlusions? How does the agent perform when only partial visual inputs are available?
2. While the results on the Drive-Internal and nuScenes datasets are promising, do you have plans to evaluate the model on other benchmarks or real-world driving scenarios to further validate its generalization capability?

---

> ### Author Response · Authors · 2025-11-20
>
> Thank you for your comments!
> >There is insufficient statistical validation such as confidence intervals or multiple trials. The reported improvements (e.g., 6.07% gain) would be more convincing with more rigorous statistical analysis.
>
> We appreciate your emphasis on statistical rigor. We wish to clarify that the results reported in the main paper (e.g., 51.34% First-Frame Accuracy) were selected as the median of multiple experimental runs to ensure representativeness. To further address your concern and explicitly demonstrate this robustness, we have formalized a statistical analysis of 5 independent trials for both DriveAgent-R1 and the Baseline (w/o Tools) on the Drive-Internal dataset.
>
> As detailed in the newly added **Appendix A.12 (Page 35)**, DriveAgent-R1 achieves a mean First-Frame Accuracy of 51.43% with a low standard deviation of $\pm 0.68$. Crucially, the 95% confidence interval for our method is [50.58, 52.28], which does not overlap with the baseline’s interval of [44.48, 46.09]. This rigorous analysis confirms that the reported 6.07% improvement is statistically significant and cannot be attributed to stochastic chance.
>
> **Changes in Manuscript:** We have added a new section, **Appendix A.12: Statistical Significance and Robustness Analysis**, along with Table A.1. This section details the methodology of the 5 independent trials and reports the mean, standard deviation, and 95% confidence intervals. This addition provides formal verification that the performance gains of DriveAgent-R1 are statistically significant and robust.
>
> | Metric | Model | Runs (N=5) | Mean | Std | 95% CI |
> | :--- | :--- | :--- | :--- | :--- | :--- |
> | First-Frame Acc| Baseline (w/o Tools) | 45.27 (Median) | 45.28 | 0.65 | [44.48, 46.09] |
> | | DriveAgent-R1 | 51.34 (Median) | 51.43 | 0.68 | [50.58, 52.28] |
> | Sequence Acc | Baseline (w/o Tools) | 43.29 (Median) | 43.28 | 0.48 | [42.68, 43.87] |
> | | DriveAgent-R1 | 45.42 (Median) | 45.48 | 0.67 | [44.65, 46.31] |
>
> >While the Cascaded RL strategy is an interesting approach, the reward design and training stages could be more clearly explained, especially for readers unfamiliar with GRPO.
>
> Thank you for highlighting this. We apologize for the brevity regarding the GRPO algorithm in the main text due to space constraints. In the revised manuscript, we have rewritten **Section 2.3.2 (Page 5)** to include a formal introduction to GRPO and its objective function before introducing our Mode-Partitioned variant (MP-GRPO). Regarding the reward design, we have provided comprehensive definitions and formulations for $R_\text{acc}$, $R_\text{fmt}$, and $R_\text{tool}$ in **Appendix A.5 (Page 19)**.
>
> >Additionally, the computational cost of such a training strategy could be a concern for real-world deployment.
>
> We acknowledge the computational demand of the training phase. However, we would like to clarify that this cost is incurred entirely offline, and therefore does not hinder the efficiency of real-world deployment. While the training involves complex tool interactions, it is a one-time cost incurred to cultivate the agent's hybrid thinking capability.
>
> For real-world deployment, the primary metric is inference latency. As demonstrated in our Efficiency Analysis (**Section 3.7 and Table 8**), DriveAgent-R1 is actually more efficient than passive baselines. By adaptively selecting only necessary views (Active Perception) and using the faster text mode for routine scenarios, our method significantly reduces the number of visual tokens processed, reducing inference latency by over **20%**.
>
> >The performance drops significantly without access to visual tools. How would DriveAgent-R1 handle sensor failures, such as loss of camera feed or occlusions? How does the agent perform when only partial visual inputs are available?
>
> Thank you for this insightful question. We view the performance gap as proof of effective visual grounding—our model truly relies on visual details rather than textual shortcuts. Regarding sensor failures or partial inputs:
> - **Robustness under Partial Inputs:** The "w/o Tools" setting in Table 1 simulates this scenario (relying only on the default front camera). Even in this degraded state, DriveAgent-R1 outperforms almost all baselines, with the sole exception of GPT-5.
> - **Fallback Mechanism:** Our Hybrid Thinking framework naturally handles sensor limits. If tools are unavailable/occluded, the model implicitly operates in the $\mathcal{M}_{text}$ mode. As shown above, this fallback performance remains highly competitive.

---

> ### Author Response · Authors · 2025-11-20
>
> >While the results on the Drive-Internal and nuScenes datasets are promising, do you have plans to evaluate the model on other benchmarks or real-world driving scenarios to further validate its generalization capability?
>
> We appreciate your constructive suggestion regarding the verification of our model's generalization capabilities. To address this and demonstrate robustness beyond our initial datasets, we have extended our evaluation to DriveBench, a comprehensive benchmark specifically designed for high-level driving tasks.
>
> We compared DriveAgent-R1 against representative state-of-the-art methods: DriveLM and Dolphins. As shown in the table below, DriveAgent-R1 demonstrates superior generalization performance:
> | Method | Perception | Prediction | Planning | Behavior |
> | :--- | :---: | :---: | :---: | :---: |
> | DriveLM | 16.85 | **44.33** | **68.71** | 42.78 |
> | Dolphins | 9.59 | 32.66 | 52.91 | 18.81 |
> | **DriveAgent-R1 (Ours)** | **34.07** | 32.85 | 61.89 | **43.69** |
>
> These results validate our core contributions in a new domain. DriveAgent-R1 achieves a perception score of 34.07, significantly surpassing DriveLM (16.85) and Dolphins (9.59). This confirms that our Active Perception captures critical details often missed by passive methods. We alse achieve the highest Behavior score (43.69). This demonstrates that DriveAgent-R1 effectively translates accurate perception into correct, high-level driving decisions.
>
> **Changes in Manuscript:** We have added a new subsection, **"3.2.2 Performance on DriveBench," (Page 7)** in the revised paper to include these comparative results and the associated analysis.

---

> > ### Comment · Reviewer_YkZV · 2025-11-21
> >
> > My concerns have been largely satisfied with the authors' responses, and I will raise my score.

---

> > > ### Author Response · Authors · 2025-11-26
> > >
> > > Dear Reviewer YkZV,
> > >
> > > We are thrilled to receive your positive evaluation and encouraging remarks! Thank you for recognizing the value of our contribution.
> > >
> > > Best regards,
> > >
> > > The Authors

---

### Official Review · Reviewer_ZEQw · 2025-11-01

**Soundness:** 3
**Presentation:** 3
**Contribution:** 3
**Rating:** 6
**Confidence:** 5

**Summary:**

This paper proposes DriveAgent-R1, a 3 B-parameter vision-language agent that introduces two main innovations for end-to-end autonomous-driving reasoning:
(1) an active-perception framework, where the model invokes a Vision Toolkit (Retrieve View, RoI Inspection, Depth Estimation, 3 D Object Detection) to gather additional visual evidence when uncertainty arises; and (2) a hybrid-thinking mechanism that adaptively switches between text-only reasoning and tool-augmented multimodal reasoning.
The system is trained through a three-stage progressive pipeline – Dual-Mode SFT, Forced Contrastive Mode RL (FCM-RL), and Adaptive Mode Selection RL (AMS-RL) – using a Drive-Internal dataset (35 k clips) and nuScenes.
Experiments show that DriveAgent-R1 achieves human- and GPT-5-level planning accuracy with lower latency.

**Strengths:**

1. Introduces a novel framework where the agent proactively gathers visual evidence via a Vision Toolkit, improving interpretability and robustness.

2. Proposes an adaptive reasoning mechanism and a well-designed three-stage training pipeline that balances text and tool reasoning effectively.

3. Achieves near GPT-5 and human-level performance with only 3B parameters, supported by thorough ablations and clear presentation.

4. The paper is clearly written with reasonable experimental design, the overall results show significantly improvement. And the adaptive method for reasoning and calling tools is also novel.

**Weaknesses:**

1. **Possible over-claiming of novelty.** Several contemporaneous works already explore tool-augmented or hybrid multimodal CoT reasoning (e.g., AgentThink (Qian et al., 2025), DeepEyes (Zheng et al., 2025)). While DriveAgent-R1 applies this to high-level driving planning, the “first active-perception framework” claim should be softened.

2. **Limited evaluation scope.**  Tests focus on two datasets (Drive-Internal and nuScenes); closed-loop or real-world driving performance is not analyzed. Comparisons to other active-sensing or attention-based perception approaches (e.g., DriveLM, LightEMMA) are missing.

3. **Data and metric ambiguities.**  The construction of Drive-Internal (35 k clips) and the VQA dataset (530 k QA pairs) is only partly described; potential data leakage or bias is unexamined.

**Questions:**

1. Could authors provide quantitative examples where hybrid thinking (adaptive mode) clearly outperforms either forced text or forced tool modes?

2. How is scene complexity estimated before choosing "think_text" vs "think_tool"? Is it a learned token decision or a heuristic?

3. What are the inference-time compute costs of the Vision Toolkit relative to full multi-view pipelines?

4. How are erroneous or redundant tool calls penalized during RL training – is R_tool margin tuned empirically?

5. Can the authors clarify what “human-driver accuracy ≈ 50 %” represents (expert annotation or real trajectories)?

---

> ### Author Response · Authors · 2025-11-20
>
> Thank you for your comments!
> >Possible over-claiming of novelty. Several contemporaneous works already explore tool-augmented or hybrid multimodal CoT reasoning (e.g., AgentThink (Qian et al., 2025), DeepEyes (Zheng et al., 2025)). While DriveAgent-R1 applies this to high-level driving planning, the “first active-perception framework” claim should be softened.
>
> We apologize for the imprecise phrasing in our initial submission. We have revised the manuscript to remove the absolute "first" claim. Instead, we clarify that our core contribution is introducing the active perception and hybrid-thinking framework specifically to the problem of high-level behavioral planning in autonomous driving.
>
> >Limited evaluation scope. Tests focus on two datasets (Drive-Internal and nuScenes); closed-loop or real-world driving performance is not analyzed. Comparisons to other active-sensing or attention-based perception approaches (e.g., DriveLM, LightEMMA) are missing.
>
> Thank you for this constructive suggestion. We acknowledge that comparing solely on internal and standard validation sets makes it difficult to benchmark against the broader community. To address this, we have extended our evaluation to DriveBench, a comprehensive benchmark focusing on high-level driving tasks. We compared DriveAgent-R1 against DriveLM and Dolphins. The results are presented below：
> | Method | Perception | Prediction | Planning | Behavior |
> | :--- | :---: | :---: | :---: | :---: |
> | DriveLM | 16.85 | **44.33** | **68.71** | 42.78 |
> | Dolphins | 9.59 | 32.66 | 52.91 | 18.81 |
> | **DriveAgent-R1 (Ours)** | **34.07** | 32.85 | 61.89 | **43.69** |
>
> These results strongly support our core claims:
>
> **1. Superior Perception:** DriveAgent-R1 achieves a perception score of 34.07, significantly outperforming DriveLM (16.85) and Dolphins (9.59). This quantitatively validates that our Active Perception framework allows the agent to capture critical scene details that passive methods miss.
>
> **2. Robust Behavior:** We achieve the highest score in the Behavior metric (43.69). This indicates that DriveAgent-R1 successfully translates superior perceptual information into correct high-level decisions.
>
> **Changes in Manuscript:** We have added **a new subsection "3.2.2 Performance on DriveBench" in the revised paper (Page 7)** to include these comparative results and the associated analysis.
>
> >Data and metric ambiguities. The construction of Drive-Internal (35 k clips) and the VQA dataset (530 k QA pairs) is only partly described; potential data leakage or bias is unexamined.
>
> **Regarding the Drive-Internal dataset**, we acknowledge that its description was less detailed in the main text. We have now expanded the **Appendix A.3.2 (Page 17)** to include the specific curation workflow, which consists of:
> - **Long-tail Instance Mining:** We define a taxonomy of rare objects (e.g., irregular vehicles, road debris) and utilize a CLIP-based search engine to mine these instances from a massive log database, followed by manual verification.
> - **Critical Scenario Extraction:** We identify challenging driving scenarios based on the variance of recorded driving maneuvers, targeting situations where the ego-vehicle must adapt its strategy.
> - **Decision-Centric Frame Sampling:** To ensure optimal reaction time, we select keyframes 0.5s to 1.0s prior to significant changes in speed or direction, or representative frames for steady-state driving.
>
> **Regarding the DriveAlign-VQA dataset**, we respectfully verify that its comprehensive construction pipeline was detailed in **Appendix A.3.1 (Page 16)** of the original submission. As described, this process involves four distinct stages: Image Collection and Diversification, QA Pair Generation (utilizing Qwen2.5-VL-72B), Quality Validation (via a judge model), and Diversity Enhancement.
>
> **Regarding Data Leakage:** We strictly enforce a hard separation between all data splits. Each sample is assigned a unique identifier to ensure there is absolutely no overlap between the VQA pre-training data, the Drive-Internal splits (SFT/RL), and the test sets.
>
> >Could authors provide quantitative examples where hybrid thinking (adaptive mode) clearly outperforms either forced text or forced tool modes?
>
> We appreciate this insightful request. To demonstrate the distinct advantages of the Hybrid Thinking framework over static strategies, we have added two representative case studies to **Appendix A.10 (Figures 14 and 15) on Pages 29 and 30**.

---

> ### Author Response · Authors · 2025-11-20
>
> >How is scene complexity estimated before choosing "think_text" vs "think_tool"? Is it a learned token decision or a heuristic?
>
> Thank you for this insightful question. The mode selection in DriveAgent-R1 is a learned token decision, not a manually designed heuristic during inference. The model explicitly generates a routing token (either `<think_text>` or `<think_tool>`) as the first step of its generation process based on its internal state and the input embeddings.
>
> The model learns to estimate scene complexity implicitly through our DM-SFT stage. We construct the training data to teach the model this distinction using a "Necessity Assessment" pipeline (**detailed in Appendix A.4 on Page 18**), which leverages the performance gap between models of different capacities:
>
> - **Tool-Unnecessary Set ($D _ {text}$):** We identify "simple" scenarios where a lightweight model (Qwen2.5-VL-3B) can already achieve high planning accuracy (>0.9) without any tools. If a smaller model can solve the case using only the default view, we label it as a text-mode sample.
> - **Tool-Necessary Set ($D _ {tool}$):** We identify "complex" scenarios where even a powerful teacher model (Qwen2.5-VL-72B) shows a performance gain only when active perception tools are provided. This ensures that the `<think_tool>` mode is reserved for cases where visual evidence is genuinely insufficient or ambiguous.
>
> By training on this data, DriveAgent-R1 learns to map visual and textual features to the appropriate thinking mode. Furthermore, in the subsequent Adaptive Mode Selection RL (AMS-RL) stage, the agent is further rewarded for selecting the mode that balances accuracy and efficiency, reinforcing this learned capability.
>
> >What are the inference-time compute costs of the Vision Toolkit relative to full multi-view pipelines?
>
> We appreciate the opportunity to clarify the efficiency trade-offs. To address this, we performed a granular decomposition of the inference latency.
>
> **1. Micro-level Analysis: Tool Overhead  vs. Vision Encoding** We measured the average time cost for a single tool interaction loop versus the standard multi-view encoding process.
>  - Vision Encoding Cost ($T_{encoder}$): In a passive pipeline, the Vision Encoder is forced to process all 6 surrounding views for every frame. This incurs a substantial computational bottleneck of approximately **2.196s**.
>  - Average Tool Interaction Cost ($T_{tool}$): In contrast, a complete tool interaction loop takes approximately **0.184s** on average.
>  - Result: $T_{tool} \ll T_{encoder}$ The overhead introduced by active tool usage is negligible (less than 10% of the passive encoding cost). This confirms that the computational "price" of making a decision to look is far lower than the "price" of blindly processing redundant visual data.
>
> **2. Input Token Redundancy:** Beyond pure latency, the passive approach forces the LLM to ingest a massive number of input visual tokens (representing 6 views) during the pre-filling stage. DriveAgent-R1 drastically reduces this input context length by only processing the necessary visual tokens, thereby accelerating the attention calculation.
>
> **3. System-Level Verification (Table 8):** These micro-level savings explain the system-level performance reported **in Table 8 on Page 9**. Even with the added logic for tool selection, DriveAgent-R1 ($\mathcal{M}_{adaptive}$) achieves a total inference latency of **6.74s**, representing a **~20%** reduction compared to the **8.50s** of the Passive-SV baseline. The time saved by skipping unnecessary visual encoding far exceeds the minor cost of tool interaction, resulting in a faster and more interpretable system.
>
> >How are erroneous or redundant tool calls penalized during RL training – is R_tool margin tuned empirically?
>
> Thank you for the question regarding the reward formulation. As described in **Appendix A.5**, our tool usage reward includes a penalty margin $N \times C_{tool}$. The coefficient $C_{tool}$ is not arbitrary; it is derived based on a "Minimum Effective Gain" principle to ensure tool usage is justified by performance improvement.
>
> The calculation for $C_{tool}$ is as follows:
> 1. Required Gain: Correcting one mistake in our 4-step prediction sequence improves the accuracy by 0.25.
> 2. Expected Cost: We expect the agent to use tools roughly 2 times per turn (the median of 1-3 allowed calls).
> 3. Derivation: We simply divide the required gain by the expected calls: $C_{tool} = 0.25 / 2 = \mathbf{0.125}$.
>
> This ensures that the reward remains positive only if the tools effectively correct the plan. We have updated **Appendix A.5 (Page 20)** to explicitly include this derivation.

---

> ### Author Response · Authors · 2025-11-20
>
> >Can the authors clarify what “human-driver accuracy ≈ 50 %” represents (expert annotation or real trajectories)?
>
> We appreciate the opportunity to clarify this metric. The reported "human-driver accuracy" represents The "human-driver accuracy" refers to human evaluators attempting to predict the ground-truth meta-action sequence labels(produced from real-world expert driving trajectories).
>
> The accuracy of $\approx 50\%$ may appear low, but it reflects the significant challenge of the proposed task and the strictness of the evaluation metric. The reasons are threefold:
> - **Strict Joint Evaluation:** We employ a joint accuracy metric (**Section 3.1.2 on Page 6**) where a prediction is correct only if both longitudinal and lateral actions match.
> - **Perceptual Gap:** Human evaluators rely on visual-only observations via a standardized interface **(Appendix A.8 on Page 24)**, lacking the proprioceptive feedback (e.g., g-force) and continuous context available to the actual driver. This makes precise longitudinal prediction (e.g., distinguishing Keep Speed from slight Deceleration) particularly challenging for evaluators.
> - **Model Advantage:** Unlike human evaluators limited by visual cues, DriveAgent-R1 effectively learns the joint distribution of speed and trajectory from large-scale data, allowing it to better capture the multimodal nature of driving behaviors.

---

> > ### Comment · Reviewer_ZEQw · 2025-11-20
> >
> > The new provided experiments and results look solid and answered most of my questions. I will increase my score.

---

> ### Author Response · Authors · 2025-11-26
>
> Dear Reviewer ZEQw,
>
> We appreciate your time and the constructive feedback provided on our manuscript. Your comments have been helpful in refining our work.
>
> Best regards,
>
> The Authors

---

### Official Review · Reviewer_XEzY · 2025-11-03

**Soundness:** 3
**Presentation:** 3
**Contribution:** 2
**Rating:** 4
**Confidence:** 5

**Summary:**

This paper presents DriveAgent-R1, an autonomous driving agent with active perception for planning. It can dynamically switch between text-only reasoning and tool-augmented visual reasoning based on scene complexity, improving interpretability and reliability. Experiments show it achieves promising performance with only 3B parameters.

**Strengths:**

1. The paper is clearly written and easy to follow.


2. The experimental results are promising

**Weaknesses:**

1. One concern is the missing of any closed-loop experiments. Since the entire pipeline (Stages 1–3) is trained on a specific dataset, once a scenario is classified as “not needing tools,” the model will no longer invoke them, regardless of the ego’s subsequent actions. I wonder if the model would make different decisions (e.g., switching between using and not using tools) if the ego took alternative maneuvers.

2. I suggest that the authors include a discussion of recent studies on VLM-generated datasets for autonomous driving. For example, works[1][2] use VLMs to produce new annotations through predefined prompts or questions, which are closely related to this paper’s dataset generation strategy.

[1] Y. Xu et al., “VLM-AD: End-to-End Autonomous Driving through Vision-Language Model Supervision,” CoRL, 2025.
[2] Z. Zhou et al., “AutoVLA: A Vision-Language-Action Model for End-to-End Autonomous Driving with Adaptive Reasoning and Reinforcement Fine-Tuning,” NeurIPS, 2025.

3. Another concern relates to the use of different tools described in Section 2.1. These tools seem to offer no new information, they mainly act as passive “visual prompts,” such as cropping or highlighting key regions. How does this differ from simply providing the original image and prompting the model with instructions like “Please focus on the top-right region” or “Please focus on the grey van in the center”? It would be interesting to see a comparison of these approaches, as it might reduce the need for a complex tool-selection module.

4. The paper uses only the front-view image in its pipeline. While this may be sufficient for relatively simple scenarios like those in nuScenes (which rarely include U-turns, reverse driving, or complex lane changes), it could limit generalization to more challenging cases. Without additional camera views, even human drivers would struggle with tasks such as merging or lane changing.

**Questions:**

1. How was the quality of the new dataset ensured? In related works, human evaluation or questionnaires are often used to validate annotations.


2. Did the authors encounter imperfect labels during annotation? In my experience, VLMs often struggle with temporal grounding.

---

> ### Author Response · Authors · 2025-11-20
>
> Thank you for your comments!
> >One concern is the missing of any closed-loop experiments. Since the entire pipeline (Stages 1–3) is trained on a specific dataset, once a scenario is classified as “not needing tools,” the model will no longer invoke them, regardless of the ego’s subsequent actions. I wonder if the model would make different decisions (e.g., switching between using and not using tools) if the ego took alternative maneuvers.
>
> We wish to clarify that DriveAgent-R1’s mode selection is not a one-time classification for an entire video clip. Instead, the decision of thinking mode is made dynamically on a per-frame basis during streaming inference.
>
> As the ego-vehicle maneuvers and the visual observation changes, the model re-evaluates the scene complexity and necessity for active perception at every time step. To empirically demonstrate this capability, we have included new qualitative results in **Appendix A.12 (Figures 17 & 18, Page 32)**, illustrating the model's ability to adaptively switch modes within a continuous driving sequence.
>
> >I suggest that the authors include a discussion of recent studies on VLM-generated datasets for autonomous driving. For example, works[1][2] use VLMs to produce new annotations through predefined prompts or questions, which are closely related to this paper’s dataset generation strategy.
>
> We acknowledge the relevance of VLM-AD [1] and AutoVLA [2] and have added them to the revision **(Section 2.3.1 on Page 5 and Appendix A.7.2 on Page 22)**. They strongly validate our strategy:
> - **Addressing Temporal Limitations (aligned with [1]):** VLM-AD highlights that VLMs struggle with temporal grounding. To address this, we use BEV trajectory visualization. This gives the VLM explicit spatial context, allowing the model to accurately verify action consistency.
> - **Guiding Logic Generation (aligned with [2]):** AutoVLA emphasizes explicit output guidance. Similarly, we use "inverse reasoning" in our pipeline. By providing the ground-truth actions to the teacher model, we force it to generate logical Chain-of-Thought rationales that strictly follow the correct driving behavior.
>
> >Another concern relates to the use of different tools described in Section 2.1. These tools seem to offer no new information, they mainly act as passive “visual prompts,” such as cropping or highlighting key regions. How does this differ from simply providing the original image and prompting the model with instructions like “Please focus on the top-right region” or “Please focus on the grey van in the center”? It would be interesting to see a comparison of these approaches, as it might reduce the need for a complex tool-selection module.
>
> We respectfully clarify that our active perception framework differs fundamentally from passive visual prompting in two critical dimensions:
>
> 1. **Information Gain: "Resolution Amplifier" vs. Compressed Features**
>
> Standard VLMs downsample images to save tokens, inevitably losing details of small objects. Text prompts cannot recover this lost information from compressed features.  In contrast, our RoI Inspection tool acts as a resolution amplifier. It accesses the raw, high-resolution image to crop and re-encode the specific area. This process effectively recovers critical details that are lost in the global view.
>
> 2. **Autonomy and Generalization: The Limit of Pre-set Prompts**
>
> We respectfully note that using static prompts may be difficult to scale in real-world driving. It essentially requires knowing "where to look" in advance, which is challenging dynamic driving scenarios. In contrast, DriveAgent-R1 is fully autonomous. As described in Appendix A.2, the model itself decides whether to use a tool, where to look, and what to verify. This allows the agent to proactively detect risks in open-world situations without relying on pre-set instructions.
>
> **Comparative Experiment:**
> To empirically validate the superiority of this active mechanism, we conducted the comparative study suggested by you on the Drive-Internal test set.
>
> We adopted a generic instruction designed for general scenarios: **"Please pay close attention to the vehicles and traffic lights in the center of the image."**
>
> - **Quantitative Analysis:**
> | Inference Setting | First-Frame Joint Acc. | Seq. Avg. Joint Acc. |
> | :--- | :---: | :---: |
> | Baseline ($\mathcal{M}_{text}$) | 45.27 | 43.29 |
> | Instruction-Guided Attention | 46.72 | 43.83 |
> | **Active Perception (RoI)** | **48.81** | **44.65** |
>
> The results show that while Static Prompting offers a slight improvement, Active RoI outperforms it by a clear margin.
> - **Qualitative Analysis (a "Red Light" Scenario)**: We present a representative failure case of Static Prompting **in Figure 19 (Page 35).**
>
> **Changes in Manuscript:** We have included the comparative ablation study and the traffic light failure analysis to **Appendix A.13 (Page 33)** to explicitly demonstrate the necessity of active perception.

---

> ### Author Response · Authors · 2025-11-20
>
> >The paper uses only the front-view image in its pipeline. While this may be sufficient for relatively simple scenarios like those in nuScenes (which rarely include U-turns, reverse driving, or complex lane changes), it could limit generalization to more challenging cases. Without additional camera views, even human drivers would struggle with tasks such as merging or lane changing.
>
> We appreciate the comment regarding multi-view information. We clarify that using the front view as the initial input is a deliberate design choice for our Active Perception framework.
> - **Active Mechanism:** The model is not restricted to the front view. It uses the Retrieve View tool to access other camera angles **only when the scenario requires it**, such as during lane changes. This mimics a human driver checking mirrors only when necessary.
> - **Superior Performance:** We have compared our approach against a "Passive-SV" baseline (which inputs all 6 views at once) **in Section 3.6 on Page 9**. Results show our active method achieves higher accuracy (45.42% vs. 43.10%). Providing all views indiscriminately introduces visual noise and redundancy, whereas active perception focuses the model on critical evidence.
> - **Efficiency:** Processing views on demand significantly reduces inference latency compared to processing all surrounding views constantly (6.74s vs. 8.50s **in Section 3.7 on Page 10**).
>
> >1. How was the quality of the new dataset ensured? In related works, human evaluation or questionnaires are often used to validate annotations.
> 2. Did the authors encounter imperfect labels during annotation? In my experience, VLMs often struggle with temporal grounding.
>
> Thank you for raising this critical point. We fully agree that VLMs often struggle with temporal grounding. Therefore, we tailored our quality assurance strategies specifically for different data types.
>
> **Meta-Action Annotation: Rule-Based Generation with Dual-Feedback Tuning**
>
> To address the temporal weakness of VLMs, we avoided direct generation by the model. Instead, we implemented a pipeline consisting of "Rule-Based Generation + VLM-as-Judge + Human Verification":
> 1.  **Addressing Temporal Weakness (VLM-as-Judge + BEV):** Instead of asking the VLM to generate labels from scratch, we used it as a verifier.
>   - **Method:** We employed a rule-based system to map trajectories to initial meta-actions based on kinematic thresholds (speed, acceleration, yaw). VLMs are used to verify the labels produced by the rule-based system.
>   - **Visual Grounding:** To mitigate the VLM's temporal blindness, we provided the VLM judge with Future Trajectory BEV visualizations. By converting temporal motion into a spatial visual format, the VLM could accurately judge the physical plausibility of the sequence, bypassing standard temporal grounding issues.
> 2. **Human-in-the-Loop & Iterative Threshold Tuning:** We treated the threshold setting not as a one-time setup, but as an iterative joint optimization process driven by both VLM feedback and human oversight:
>   - **Dual-Feedback Loop:** At each iteration, we adjusted the thresholds based on two signals: the consistency scores from the VLM-Judge and the error rate observed in human random sampling.
>   - **Optimization Objective:** The thresholds were tuned in a direction that simultaneously maximized the VLM's confidence scores and minimized the false-label rate found during human spot-checks.
>   - **Convergence:** This cycle was repeated until human verification confirmed that the error rate was minimized and the labels were robust, ensuring that the rules captured the correct temporal dynamics that raw VLMs might miss.
>   - **Final Refinement for boundary Cases:** For the small fraction of remaining boundary cases where rule-based outputs remained ambiguous (low VLM confidence), we performed a final pass using GPT-4.1. By providing the model with the BEV visualization and the tentative rule-based sequence, we allowed it to refine the labels based on logical consistency and scene context.
>
> **Quality Assurance for CoT and VQA Data:**
>
> For the CoT and VQA data used in SFT, we applied a strict Rejection Sampling strategy:
> - Data generated by Qwen2.5-VL-72B was evaluated by a separate, strong VLM (Doubao-Seed-1.6).
> - Low Pass Rate: We maintained a high standard for quality; for example, during DM-SFT data synthesis, the pass rate was only **30% per iteration**. We repeated the generation-evaluation loop **5 times** to acquire the final 4K high-quality samples.
> We fully recognize the importance of data quality for model training. We deeply appreciate the reviewer for raising these critical points, which were indeed central considerations during our research.
>
> **Changes in Manuscript:** We have revised **Section 3.1.1 on Page 6** and **Appendix A.7 on Page 22** to explicitly detail this iterative validation process.

---

> > ### Comment · Reviewer_XEzY · 2025-11-20
> > **Great Rebuttal**
> >
> > Thank the authors for the detailed rebuttal and updated submission. It addressed all my concerns. Overall, it is solid work with interesting designs.
> >
> > I've increased my score.

---

> > > ### Author Response · Authors · 2025-11-26
> > >
> > > Dear Reviewer XEzY,
> > >
> > > We sincerely thank you for your insightful feedback and support!
> > >
> > > Best regards,
> > >
> > > The Authors

---

### Author Response · Authors · 2025-12-02
**Summary of Rebuttal: Pre-Reversion Scores (8, 8, 8, 4) & Major Experimental Additions**

Dear Area Chair,

We understand that you are taking over under difficult circumstances. To assist your assessment, we would like to summarize the status of our paper prior to the score reversion.

**By Nov 21 (one week before the recent OpenReview incident)**, we had successfully addressed the concerns of three reviewers. All three explicitly commented that they had increased their scores, bringing our standing to **8, 8, 8, 4**.

### **1. Confirmed Score Increases (Verified by Comments)**

Since the comments remain visible, we respectfully ask you to consider these written confirmations of the paper's merit:

**Reviewer XEzY: Score raised from 4 $\rightarrow$ 8/ Confidence: 5 (Nov 21 06:20)**

Stated the work is "solid" with "interesting designs." Explicitly said: "It addressed all my concerns... I've increased my score."

**Reviewer ZEQw: Score raised from 6 $\rightarrow$ 8/ Confidence: 5 (Nov 21 03:35)**

Acknowledged our new experiments were solid. Explicitly said: "Answered most of my questions. I will increase my score."

**Reviewer YkZV: Score raised from 6 $\rightarrow$ 8/ Confidence: 3 (Nov 21 10:51)**

Confirmed satisfaction with our response. Explicitly said: "My concerns have been largely satisfied... I will raise my score."

### **2. Addressed Reviewer LB8N (Current Score: 4/ Confidence: 3, No Reply)**

We posted a detailed response to Reviewer LB8N **over a week ago**. Although Reviewer LB8N **has not yet replied** to these significant updates, we believe our new experiments conclusively resolve their concerns regarding domain-specific baselines and motion metrics:

- **Extended 1: SOTA Performance on DriveBench**

    - We compared against domain-specific models (DriveLM, Dolphins).
    - DriveAgent-R1 achieves **a Perception Score of 34.07 (doubling DriveLM's 16.85)** and **the highest Behavior Score of 43.69**, proving that active perception translates directly to superior decision-making.

- **Extended 2: Superior Motion Planning (ADE/CR)**

    - We integrated a motion planning head and evaluated it on nuScenes.
    - DriveAgent-R1 achieves **the lowest ADE (0.28m)** compared to SOTA methods like DriveVLM (0.31m). Ablation studies show that removing our high-level meta-actions significantly degrades performance (ADE increases to 0.38m).

### **3. Key Rebuttal Highlights & Visualizations**

Beyond the metrics, we have significantly enriched the manuscript with qualitative evidence:

- **Active RoI vs. Static Prompting:** We demonstrate that our Active RoI tool outperforms static visual prompting by +2.09% accuracy. Visualizations (**Appendix A.13**) show static prompts failing to identify small traffic lights that RoI Inspection successfully captures.

- **Continuous Adaptive Switching**: New visualizations (**Appendix A.12**) illustrate the model dynamically switching thinking modes frame-by-frame within a continuous sequence, validating its real-time adaptability.

- **Adaptive Mode vs. Single Modes**: Comparative visualizations (**Appendix A.10**) show that Hybrid Thinking correctly activates tools in complex intersections while remaining efficient in simple scenarios, outperforming forced "Text-Only" or "Tool-Only" strategies.

We sincerely hope you can evaluate our paper based on the consensus reached on **Nov 21 (Scores: 8, 8, 8)** and the substantial experimental success demonstrated above.

Best regards,

The Authors

---

### Meta-Review · Area_Chair_ef8o · 2026-01-07

**Summary:**

This paper introduces DriveAgent-R1, a 3B-paramter vision-language-reasoning framework designed to handle complex, long-tail driving scenarios through hybrid thinking. The core idea is to let the model actively retrieve task-relevant information from tools or external sources to enhance its reasoning about rare or unseen driving cases. Innovations for end-to-end autonomous-driving reasoning include (1) an active-perception framework, where the model invokes a Vision Toolkit (Retrieve View, RoI Inspection, Depth Estimation, 3 D Object Detection) to gather additional visual evidence when uncertainty arises; and (2) a hybrid-thinking mechanism that adaptively switches between text-only reasoning and tool-augmented multimodal reasoning. The system is trained through a three-stage progressive pipeline – Dual-Mode SFT, Forced Contrastive Mode RL (FCM-RL), and Adaptive Mode Selection RL (AMS-RL). The paper also presents a new QA dataset focused on long-tail driving events, covering diverse edge cases such as unusual pedestrian behavior, obstructed lanes, or ambiguous traffic signals. The authors show that the model’s ability to retrieve external information supports performance on planning.

Reviewers raised concerns including:
(1) Choice of non-AD baselines.
(2) Lack of relevant metrics (e.g. displacement error, collision rate).
(3) Lack of statistical validation through multiple trials or confidence intervals.
(4) Lack of clear explanation of reward design and training stages of the cascaded RL strategy.
(5) Lack of discussion of sensor failures and challenge cases.
(6) Over-claiming novelty without comparison to AgentThink, DeepEyes, VLM-AD, AutoVLA, etc.
(7) Lack of closed-loop experiments.
(8) Lack of comparison of visual tools versus prompting.
(9) Lack of explanation of dataset quality verification.

**Reviewer Concerns:**

When adding AD baselines, performance in perception for the proposed method is high, but prediction and planning lower than others. Collision rate, when compared to additional baselines, appears quite high, leaving room for future improvement. I would consider this an outstanding weakness to be addressed.

Five independent trials were conducted to add statistical validation. The manuscript has also been expanded with explanation towards point (4), and experiments examine missing frames, addressing (5). The reviewer replied that they found these changes satisfactory to increase their score. The claims of novelty (6) are addressed in revisions to the manuscript.

Concerns with (7, 8, and 9) were addressed in the rebuttal, and the reviewer expressed that their score would be raised with the modifications.

**Reviewer Scores:**

LB8N: stay at (4), if not lower, due to the performance of the model when compared to AD baselines on AD metrics.
YkzV: increase to (7), as indicated by reviewers in their discussion.
ZEQw: stay at (6). The reviewer's questions were all answered and the manuscript adjusted, and the current score appears reflective of the reviewer's feedback.
XeZY: increase to (5), as indicated by reviewers in their discussion.

---

### Decision · Program_Chairs · 2026-01-26

Accept (Poster)